# A Two-Component Parameterization of Marine Ice Nucleating Particles Based on Seawater Biology and Sea Spray Aerosol Measurements in the Mediterranean Sea

Jonathan V. Trueblood[1], Alesia Nicosia[1], Anja Engel[2], Birthe Zäncker[2], Matteo Rinaldi[3], Evelyn Freney[1], Melilotus Thyssen[4], Ingrid Obernosterer[5], Julie Dinasquet[5,6], Franco Belosi[3], Antonio Tovar-Sánchez[7], Araceli Rodriguez-Romero[7], Gianni Santachiara[3], Cécile Guieu[5], and Karine Sellegri[1]

[1] Université Clermont Auvergne, CNRS, Laboratoire de Météorologie Physique (LaMP) F-63000 Clermont-Ferrand, France
[2] GEOMAR, Helmholtz Centre for Ocean Research Kiel, 24105 Kiel, Germany
[3] Institute of Atmospheric Sciences and Climate, National Research Council, 40129 Bologna, Italy
[4] Mediterranean Institute of Oceanography, 163 avenue de Luminy, Marseille, France
[5] CNRS, Sorbonne Université, Laboratoire d'Océanographie de Villefranche, UMR7093, Villefranche-sur-Mer
[6] Marine Biology Research Division, Scripps Institution of Oceanography, 92037 La Jolla, US
[7] Department of Ecology and Coastal Management, Institute of Marine Sciences of Andalusia (ICMAN-CSIC), 07190 Puerto Real, Spain

*Correspondence to:* K.Sellegri@opgc.cnrs.fr

**Abstract.** Ice nucleating particles (INP) have a large impact on the climate-relevant properties of clouds over the oceans. Studies have shown that sea spray aerosols (SSA), produced upon bursting of bubbles at the ocean surface, can be an important source of marine INP, particularly during periods of enhanced biological productivity. Recent mesocosm experiments using natural seawater spiked with nutrients have revealed that marine INP are derived from two separate classes of organic matter in SSA. Despite this finding, existing parameterizations for marine INP abundance are based solely on single variables such as SSA organic carbon (OC) or SSA surface area, which may mask specific trends in the separate classes of INP. The goal of this paper is to improve the understanding of the connection between ocean biology and marine INP abundance by reporting results from a field study and proposing a new parameterization of marine INP that accounts for the two associated classes of organic matter. The PEACETIME cruise took place from May 10 to June 10, 2017 in the Mediterranean Sea. Throughout the cruise, INP concentrations in the surface microlayer ($INP_{SML}$) and in SSA ($INP_{SSA}$) produced using a plunging aquarium apparatus were continuously monitored while surface seawater (SSW) and SML biological properties were measured in parallel. The organic content of artificially generated SSA was also evaluated. $INP_{SML}$ and $INP_{SSA}$ concentrations were found lower than in the literature, presumably due to the oligotrophic nature of the Mediterranean Sea. A dust wet deposition event that occurred during the cruise increased the INP concentrations measured in the SML by an order of magnitude, in line with increases of iron in the SML and bacterial abundances. Increases of $INP_{SSA}$ were not observed until after a delay of three days compared to increases in the SML, and are likely a result of a strong influence of bulk SSW INP for the temperatures investigated (T=-18°C for SSA, T=-15°C for SSW). Results confirmed that $INP_{SSA}$ are divided into two classes depending on their associated organic matter. Here we find that warm (T ≥ -22°C) $INP_{SSA}$ concentrations are correlated with water soluble organic matter (WSOC) in the SSA, but also to SSW parameters ($POC_{SSW}$ and $INP_{SSW,-16C}$) while cold $INP_{SSA}$ (T < -22°C) are correlated with SSA water-insoluble organic carbon (WIOC), and SML dissolved organic carbon (DOC) concentration. A relationship was also found between cold $INP_{SSA}$ and SSW nano- and micro-phytoplankton cell abundances, indicating that these species might be a source of water insoluble organic matter with surfactant properties and specific IN activities. Guided by these results, we formulated and tested multiple parameterizations for the abundance of INP in marine SSA, including a single component model based on $POC_{SSW}$ and a two-component model based on SSA WIOC and OC. We also altered two previous models based on SSA surface area and $OC_{SSA}$ content to account for oligotrophy of the Mediterranean Sea. We then compared these formulations with the previous models. These new parameterizations should improve attempts to incorporate marine INP emissions into numerical models.

## 1 Introduction

Ice nucleating particles are a subset of aerosol particles that are required for the heterogeneous nucleation of ice particles in the atmosphere. While extremely rare (Rogers et al., 1998), INP greatly control the ice content of clouds, which is crucial to a range of climate-relevant characteristics including precipitation onset, lifetime, and radiative forcing (Verheggen et al., 2007). Despite their importance, the knowledge of INP sources and concentrations, particularly in marine regions, remains low as evidenced by the large uncertainties in modelled radiative properties of clouds (McCoy et al., 2015; McCoy et al., 2016; Franklin et al., 2013).

While the ice nucleating (IN) ability of marine SSA particles is less efficient than their terrestrial counterparts (DeMott et al., 2016), modelling studies have shown that marine INP are of particular importance in part due to the lack of other INP sources in such remote regions (Burrows et al., 2013; Vergara-Temprado et al., 2017). For this reason, recent studies have been conducted to better understand which SSA particles contribute to the marine INP population as well as the relationship between SSA emission and ecosystem productivity. Results from these studies suggest that the IN ability of SSA is linked to the biological productivity of source waters, with higher productivity leading to greater IN activity (DeMott et al., 2016; Bigg, 1973; Schnell and Vali, 1976). For example, it has been shown that both the cell surface and organic exudate of the marine diatom *Thalassiosira pseudonana* can promote freezing at conditions relevant to mixed-phase clouds (Knopf et al., 2011; Wilson et al., 2015). More recently, mesocosm studies on phytoplankton blooms using two separate in-lab SSA-generation techniques have furthered the understanding of the connection between ocean biology and the IN activity of SSA (McCluskey et al., 2017). In-depth chemical analysis of the artificially generated SSA during this set of experiments has revealed marine INP may be related to two classes of organic matter: a regularly occurring surface-active molecule type related to DOC and long-chain fatty acids, and an episodic heat-labile microbially-derived type (McCluskey et al., 2018b).

As the understanding of the connection between ocean biology and marine INP has improved, parameterizations for predicting marine INP abundance using readily available ocean parameters have been proposed. Wilson and co-authors (Wilson et al., 2015) identified a temperature-dependent relationship between TOC and ice nucleating entities (INE) number concentrations in the SML from samples collected in the North Atlantic and Arctic ocean basins. They then extended this relationship from the ocean to the atmosphere to predict the abundance of INP in SSA based on model estimates of marine organic carbon aerosol concentrations. The parameterization was tested for the first time on field measurements of marine aerosol over the North Atlantic at Mace Head and was found to overestimate INP abundance in pristine marine aerosol by a factor of 4 to 100 at -15°C and -20°C (McCluskey et al., 2018c). In the same study, a new parameterization based on SSA surface area and temperature was proposed (McCluskey et al., 2018c). However, this parameterization did not incorporate the recently observed heat labile organic INPs. Most recently, this parameterization was compared with observations of INP over the Southern Ocean, showing reasonable agreement between predictions and observations at -25°C (McCluskey et al., 2019).

Despite the recent progress made in the understanding of marine INP, there remains much room for improvement. To date, previous parameterizations have only been tested in the two field studies mentioned in the previous paragraph, underscoring the need for more real-world observations. Furthermore, the field studies conducted so far have taken place in regions of the ocean where biological productivity is high (i.e., North Atlantic and Southern Ocean). As modelling work has shown that the link between ocean biology and SSA organic content properties in oligotrophic waters differs from those in highly productive regions (Burrows et al., 2014) there is need for more measurements in waters with low primary productivity. Finally, despite the finding that marine INP may exist as two separate populations, no model has yet been proposed to account for this.

This paper addresses the current gaps in the knowledge of marine INP by 1) testing existing parameterizations of INP on a new set of field measurements by extending the current inventory of field measurements beyond eutrophic waters to more oligotrophic regions for the first time 2) improving the understanding of how INP in the SML and SSA are linked to both

seawater biological and SSA organic properties and 3) proposing a new parameterization based on the two-component nature of INP. Here we present results from the ProcEss studies at the Air-sEa Interface after dust deposition in the Mediterranean Sea (PEACETIME) cruise. The cruise took place in the central and western Mediterranean Sea from May 10 - June 10, 2017. Observations of INP concentrations both in the SML and SSA were compared with a suite of surface seawater, surface microlayer, and SSA properties to better determine how INP concentrations related to biology.

## 2 Methods

In the frame of the PEACETIME project (http://peacetime-project.org/), an oceanographic campaign took place aboard the French research vessel (R/V) 'Pourquoi Pas?' between May 10-June 10, 2017 with the purpose of investigating the processes that occur at the air-sea interface in the Mediterranean Sea. The cruise started in La Seyne, France and travelled in a clockwise fashion between 35˚ to 42˚ latitude and 0˚ to 20˚ longitude. The observations and process studies performed on board both in the whole water column and the atmosphere are described elsewhere (Freney et al., 2020). Here, we focus on the measurements conducted to describe the SML, SSW, and aerosol properties.

### 2.1 Surface Seawater (SSW)

SSW properties presented here were obtained from sampling at depths of 20 cm and 5 m. First, 21 parameters including various chemical properties, microbial assemblages, hydrological properties, and optical properties were monitored using the ship's underway system that continuously collected seawater at 5 m under the ship using a large peristaltic pump (Verder VF40 with EPDM hose). These measurements included counts of specific microbial classes (e.g., *Synechococcus*, *Prochlorococcus*, picoeukaryotes, nanoeukaryotes, microphytoplankton, high phycoerythin containing cells, coccolithophores, cryptophytes), as well as seawater biovolume, chlorophyll-*a* (chl-*a*), and POC concentrations. Chl-*a* was determined from the particulate absorption spectrum line-height at 676 nm after adjusting to PEACETIME chl-*a* from high performance liquid chromatography (HPLC). POC was estimated from the particulate attenuation at 660 nm using an empirical relationship specific to PEACETIME (POC = 1405.1 x $c_p$(660) – 52.4). For enumeration of phytoplankton cells, an automated Cytosense flow cytometer (Cytobuoy, NL) operating at a time resolution of one-hour was connected to the continuous underway seawater system. Particles were carried in a laminar flow filtered seawater sheath fluid and subsequently detected with forward scatter and sideward scatter as well as fluorescence in the red (FLR > 652 nm) and orange (FLO 552-652 nm). Distinction between highly concentrated picophytoplankton and cyanobacteria groups and lower concentrated nano- and microphytoplankton was accomplished using two trigger levels (trigger level FLR 7.34 mV, sampling speed of 4 $mm^3 s^{-1}$ analysing $0.65 \pm 0.18$ $cm^3$ and trigger level FLR 14.87 mV at a speed of 8 $mm^3 s^{-1}$ analysing $3.57 \pm 0.97$ $cm^3$).

The second set of SSW measurements were made on seawater collected at ~20 cm depth from a pneumatic boat that was periodically deployed at a distance of 2 km from the R/V to avoid contamination. The SSW was manually collected using acid cleaned borosilicate bottles. From these discrete samples, microbial composition and cell abundance of the SSW was monitored as described in a companion paper (Tovar-Sanchez et al., 2019). Measurements included heterotrophic bacteria counts, high nucleic acid and low nucleic acid bacteria (HNA and LNA bacteria, respectively), total non-cyanobacteria like cells (NCBL), cyanobacteria like cells (CBL), and total phytoplankton concentration (NCBL+CBL). These were further segregated into size classes of small, medium, and large which roughly correspond to the pico-, nano-, and micro- size classifications from the underway measurements. Trace metals (i.e., Cd, Co, Cu, Fe, Ni, Mo, V, Zn, Pb) were analysed by inductively coupled plasma mass spectrometry, although here we only report on Fe. Finally DOC and marine gel-like particles, including abundance of transparent exopolymer particles (TEP) and Coomassie stainable particles (CSP) were also measured as described in literature (Engel, 2009).

**2.2 Surface Microlayer**

At the same time SSW samples were manually collected on the pneumatic boat, SML samples were also collected using a glass plate sampling method which has been previously described in the literature (Tovar-Sanchez et al., 2019). The glass plate was cleaned overnight with acid and rinsed with ultrapure Milli-Q water. Roughly 100 dips of the glass were conducted to collect 500 mL of SML water into 0.5 L acid cleaned low-density polyethylene plastic bottles. The samples were then acidified on board to pH<2 with ultrapure-grade hydrochloric acid in a class-100 HEPA laminar flow hood. The same measurements done for the SSW samples (see above, Section 2.1) were then made on the SML samples. Enrichment factor was calculated for relevant properties as the ratio of SML to SSW:

$$EF = \frac{SML}{SSW}$$

In addition to biological measurements, concentrations of immersion freezing mode INP in SML samples (and a small number of SSW samples, n=4) were measured between May 22 - June 7 using an offline method described previously (Stopelli et al., 2014). Briefly, prior to acidification of the SML samples, additional aliquots were separated and stored in Corning Falcon 15 mL conical tubes and frozen at -20°C until analysis. Before INP measurement, each aliquot was gradually defrosted and distributed into an array of 26 Eppendorf tubes filled up to 200 µL. The array was then immersed inside an LED based Ice Nuclei Detection Apparatus (LINDA) and the number of ice nucleating particles per liter (INP/L) of SML water was following the method described in Stopelli et al. (2014) which was originally formulated by Vali (1971):

$$\frac{INP}{volume} = \frac{\ln(N_{total}) - \ln(N_{unfrozen})}{V_{tube}}$$

where $N_{total}$ is the total number of tubes, $N_{unfrozen}$ the total number of unfrozen tubes, and $V_{tube}$ the volume of sample in each tube. The number of unfrozen tubes is calculated by first blank correcting the number of frozen tubes, and then subtracting that value from the total number of tubes. We calculated uncertainty as the binomial proportion confidence interval (95%) using the Wilson score interval. Samples were not corrected for salinity in this study.

**2.3 Artificially Generated Sea Spray Aerosol**

Sea spray aerosols were generated using a sea spray generation apparatus which has been described previously (Schwier et al., 2015; Schwier et al., 2017). The characteristics of the setup were selected to mimic Fuentes et al. (2010). These parameters (water flow rates, plunging water depth, etc.) have been shown to mimic well nascent SSA. The apparatus consists of a 10 L glass tank with a plunging jet system. A continuous flow of seawater collected at 5 m depth using the ship's underway seawater circulating system (described above) was supplied to the apparatus. Particle free air was passed perpendicular to the water surface at a height of 1 cm to send a constant airflow across the surface of the water. Aerosols were then either dried with a 1 m long silica dryer for online instrumentation (see Section 2.3.3), with a 30 cm silica gel dryer cascade impactor sampling with subsequent chemical analysis, or were sampled directly from the sea spray generator onto filters for INP analysis.

**2.3.1 Offline PM1 Filter Analysis**

Aerosol particles were also sampled onto PM1 quartz fiber filters mounted on a 4-stage cascade impactor (10 LPM) on a daily basis (24-hour duration). Samples were then extracted in Milli-Q water by sonication for 30 minutes for the analysis of water-soluble components. Main inorganic ion abundance (i.e., $SO_4^{2-}$, $NO_3^-$, $NH_4^+$, $Na^+$, $Cl^-$, $K^+$, $Mg^{2+}$, $Ca^{2+}$) was analysed via ion chromatography. An IonPac CS16 3x 250 mm Dionex separation column with gradient methanesulfonic acid elution was used for cations, while an IonPac AS11 2 x 250 mm Dionex column with gradient potassium hydroxide elution was used for anions. Water soluble organic carbon (WSOC) and water insoluble organic carbon (WIOC) were also determined. WSOC was measured after water extraction using a high-temperature catalytic oxidation instrument (Shimadzu; TOC 5000 A). Total

organic carbon (which we now refer to as OC), was measured using a Multi N/C 2100 elemental analyzer (Analytik Jena, Germany) with a furnace solids module. The analysis was performed on an 8 mm diameter filter punch, pre-treated with 40 µL of $H_3PO_4$ (20% v/v) to remove contributions from inorganic carbon. WIOC was determined as the difference between OC and WSOC. Finally, we calculated organic mass fraction of SSA (OMSS) by taking the ratio of OM/(OM+SeaSalt), where OM is the sum of WSOM and WIOM, calculated as WSOM = WSOC x 1.8 and WIOM = WIOC * 1.4 and SeaSalt is the sum of inorganic ion abundance as determined above.

**2.3.2 INP**

INP concentrations were determined from filter-based samples of total suspended particles over a 24-hour duration daily or from the average of two filters (day and night). The concentration of INP in the SSA was determined for the condensation freezing mode using a Dynamic Filter Processing Chamber (DFPC), which has been used in multiple previous studies and found to agree well with other INP monitoring instruments (DeMott et al., 2018; Hiranuma et al., 2019; McCluskey et al., 2018c). A full description of the instrument can be found in the literature (DeMott et al., 2018). Briefly, bulk SSA formed using the plunging aquarium apparatus were impacted onto 47 mm nitrocellulose filters which were then placed on a metal plate coated with a smooth surface of Vaseline. Air entered the chamber and was sent through a cooling coil allowing it to become saturated with respect to water. Different supersaturations with respect to ice and liquid water can be obtained by controlling the temperatures of the filter and the air flowing across the filter. Filter air temperature combinations were set three different ways, all resulting in a supersaturation with respect to liquid water of 1.02. The filter temperatures were -18, -22, and -25°C (-15.9, -19.6, and -22.3°C for air temperature). Under these conditions, condensation freezing is expected to be the dominant freezing mode for INP. It has been reported (Vali et al., 2015) that condensation freezing and immersion freezing are not distinguishable from one another. Filters were processed inside the DFPC for 15 minutes and monitored for formation of ice crystals upon activation of INPs. Based on sampling time and flow rate, the number of INP/volume were calculated. We report an uncertainty of ±30% based on previous reports of the DFPC (DeMott et al., 2018).

**2.3.3 Size Distribution Measurements**

Particle size distribution and number concentrations of aerosols generated with the plunging apparatus were monitored using a custom-made differential mobility particle sizer (DMPS) preceded by a 1-micron size-cut impactor and X-ray neutralizer (TSI Inc.). Total counts from the DMPS system were checked using a condensation particle counter (CPC, TSI3010). Using the DMPS, a total of 25 size bins ranging between 10-500 nm (dry particle electrical mobility diameter) were scanned over a 10-minute time period. We then averaged the size distributions across each DFPC sampling period. For the purpose of the present study, surface area of SSA particles were calculated from the number size distributions by assuming spherical particles. Theoretical calculation of the number and surface area distributions for particles between .5-10 µm was also carried out. The fit from our observed number size distributions from modes 1-4 agreed well with the fit of a sea spray aerosol source function consisting 5 lognormal modes based on in-situ particle number concentration measurements at Mace Head and open-ocean eddy correlation flux measurements from the Eastern Atlantic (Table S1) (Ovadnevaite et al., 2014). We took the ratio of mode 5 to mode 3 from this parameterization and applied it to our fit to calculate a fifth mode accounting for particles ranging in size between 500 nm and 10 µm.

    **3 Results**

**3.1 INP in the Seawater and SSA**

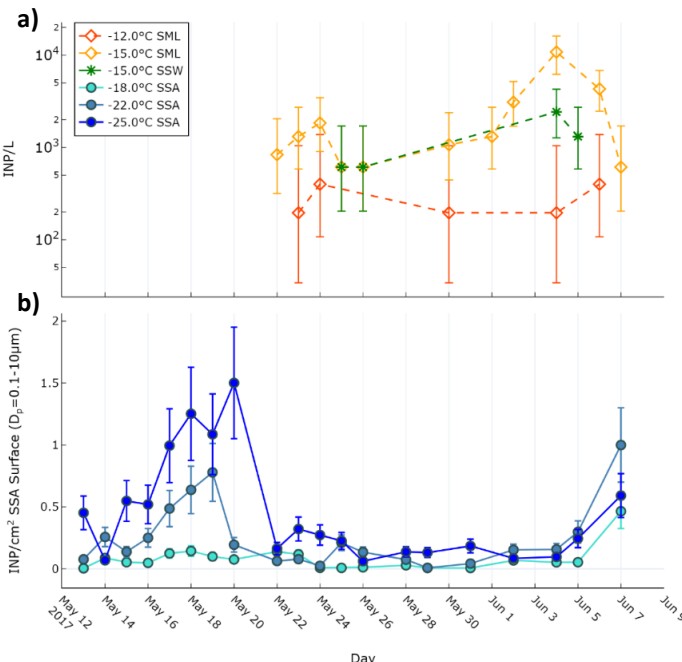

**Figure 1. a) INP concentrations observed during the PEACETIME cruise in the SML and SSW as measured using the LINDA**
**instrument. Error bars represent the binomial proportion confidence interval (95%) using the Wilson score interval. b) INP$_{SSA}$**
**concentrations as observed by the DFPC normalized by SSA surface area. Error bars represent ±30% uncertainty of the DFPC**
**instrument, as cited previously (DeMott et al., 2018).**
Ice nucleating particle characteristics were determined for the SSW, SML, and SSA. Figure 1a shows the concentration
of INP in the SML (INP$_{SML}$) at two different temperatures (-12°C, -15°C) and in the SSW (INP$_{SSW}$) at -15°C as determined
using the LINDA instrument. An initial increase occurred on May 24 ($1.8 \times 10^3$ INP/L at T=-15°C) relative to May 22 which
was then followed by a further increase on June 4 ($1.1 \times 10^4$ INP/L at T=-15°C). The enhancement on June 4 occurred on the
same day as a dust deposition event which led to an enrichment of iron in the SML relative to the underlying water (see Section
3.2). While only four SSW samples were analysed for INP concentrations, they exhibited similar concentrations and trends to
those seen in the SML, with an observed maximum on June 4 ($2.4 \times 10^3$ INP/L at T=-15.0°C). Based on these four samples, no
significant enrichment of INP was observed in the SML compared to SSW, except during the dust deposition event when the
SML concentration was enriched by a factor 4.5.
Figure 1b shows the concentration of ice nucleating particles in SSA (INP$_{SSA}$) normalized by particle surface area for
particles with diameters smaller than 10 µm ($0.1 < D_p < 10$ µm) at three different temperatures as observed by the DFPC. It
should be noted that INP$_{SSA}$ measurements were conducted at colder temperatures than for the INP$_{SML}$ measurements due to
differences between the LINDA and DFPC instruments. In general, the highest concentrations of INP$_{SSA}$ were observed at the
beginning of the voyage, with an initial value of 0.45 INP$_{SSA,-25C}$/cm² observed on May 13, increasing to a maximum observed
value of 1.5 INP$_{SSA,-25C}$/cm² on May 20. After May 20, a considerable drop in INP$_{SSA,-25C}$ concentrations was observed.
Concentrations remained at low, albeit with slight fluctuations, before increasing again to 0.59 INP$_{SSA,-25C}$/cm² on June 7. It is
also worth noting that the highest concentrations of INP active at -18°C (INP$_{SSA,-18C}$/cm²) were observed on this day. The
increase of INP concentrations around the time of the dust deposition event in early June is similar to the trend observed for
seawater INP, albeit with a lag of at least one day (no observations of INP$_{SSA}$ were made on June 6).

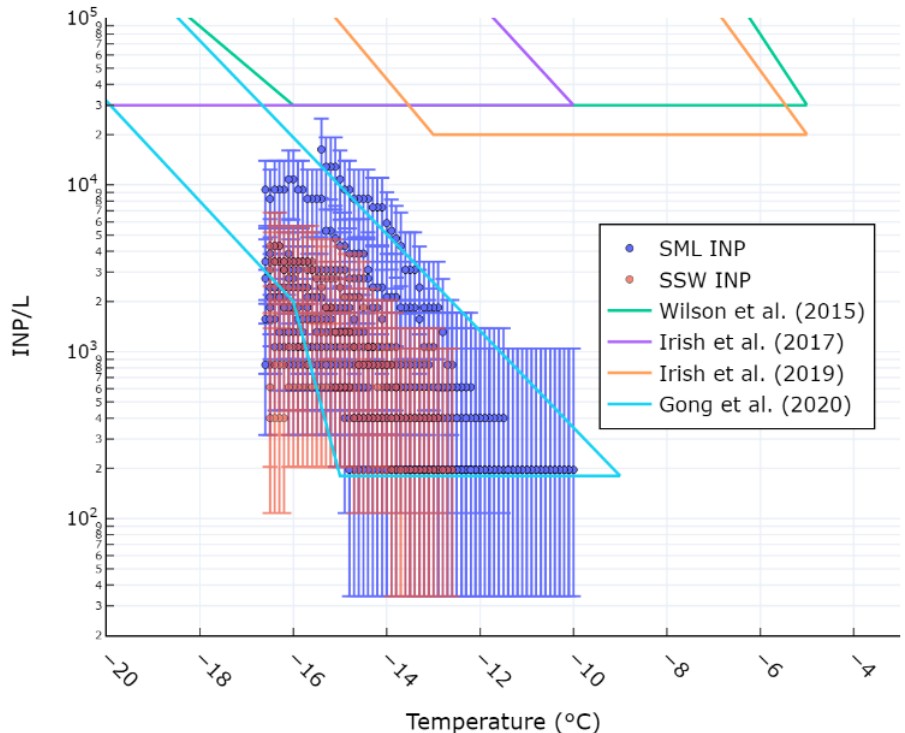

Figure 2. Comparison of observed SSW (blue markers) and SML (red markers) INP concentrations with previous studies. Error bars represent the binomial proportion confidence interval (95%) using the Wilson score interval.

Figure 2 shows the comparison of observed INP concentrations at various temperatures in the SML and SSW with those reported in previous studies. The concentrations we report here are lower than those from Arctic seawater samples reported by Irish et al. (2017; 2019) and from Arctic and North Atlantic seawater samples reported in Wilson et al. (2015). The difference can likely be attributed to the fact that eutrophic Arctic and North Atlantic seawater is more biologically active than the oligotrophic Mediterranean Sea. Our values agree well with those reported by Gong et al. (2020) who calculated INP concentrations in mid-latitude seawater off the coast of Cabo Verde. The authors of that study also posited that the low INP concentrations relative to Irish et al. (2017; 2019) and Wilson et al. (2015) was due to the lower biological activity of the oligotrophic seawater near Cabo Verde.

Figure 3 compares the INP per cm$^2$ of SSA surface during the PEACETIME cruise with values reported in previous studies. DeMott et al. (2016) reported INP concentrations from ambient measurements over the Caribbean, Arctic, Pacific, and Bering Sea. DeMott et al. (2016) and McCluskey et al. (2017) both reported INP concentrations from separate experiments in which SSA was artificially generated using nutrient-spiked seawater collected off the Scripps Institute of Oceanography (SIO) Pier with either a marine aerosol reference tank (MART) or an indoor waveflume. McCluskey et al. (2018a) reported ambient INP concentrations measured over the Southern Ocean. Finally, Gong et al. (2020) reported INP concentrations as measured off the coast of Cabo Verde. Our observed values are below those in all studies cited. The differences in our values compared to those in the literature can be attributed to a number of factors, including differences in trophic state of source waters, influences from terrestrial sources, and differences in INP analysis instruments. For example, Gong et al. (2020) state that most INPs observed in their study were from dust particles, rather than sea spray. Gong et al. (2020) also calculated a theoretical INP concentration based on the ratio of NaCl mass to INP in air and seawater (not shown in Figure 3). When we perform the same calculation using observed NaCl values in SSA and salinity measurements of underway seawater, we find that the INP/NaCl in the SSA is ~3000 times higher than in the SML. This is an important conclusion and points to the need for caution when using the Gong et al. (2020) approach for calculating a contribution of SSA-derived INP in ambient air aerosols in future studies. We note that studies in which seawater has been spiked with nutrients (McCluskey et al., 2017; McCluskey et al., 2018b; DeMott et al., 2016) are expected to have higher levels of biological activity than those observed in the Mediterranean and other oligotrophic regions. Since the departure of the PEACETIME INP in SML and SSW to the literature values is of the

same order of magnitude as the departure of the PEACETIME INP in SSA to the literature, it is reasonable to attribute the low
INP$_{SSA}$ values to the oligotrophic nature of the Mediterranean seawater.

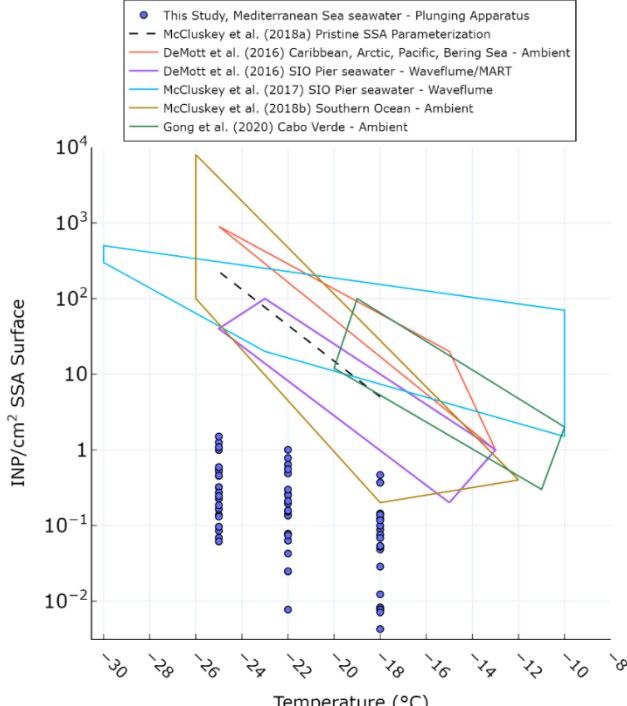

**Figure 3. INP/cm$^2$ SSA surface (0.1 < D$_p$ < 10 µm) at various temperatures as measured by the DFPC during the PEACETIME**
**cruise (blue circles) compared with values reported in the literature. SSA were generated by continuously passing seawater from**
**the ship's underway system into a plunging apparatus. Error bars are not shown as the uncertainty is smaller than the data points.**
**3.2 Correlations between INP and Biogeochemical Conditions**
As described in the methods section, various seawater biogeochemical properties were monitored throughout the voyage
for the SSW and SML. Plots of selected continuous measurements from the R/V's underway sampling system and discrete
measurements from the pneumatic boat of relevant biogeochemical values are found in the supporting information (Figure S1
and Figure S2, respectively). Biogeochemical properties are described in more detail in our companion papers (Freney et al.,
2020; Tovar-Sanchez et al., 2019) and seawater gel properties will be discussed in an upcoming paper. Here, we present a
broad summary of observed conditions.
In general, surface waters were characterized by oligotrophic conditions as expected for the season. Bacteria
concentrations ranged between $2x10^5$ and $7x10^5$ cells/mL in the SSW and were greatest at the start and end periods of the
voyage. NCBL abundance followed a similar trend and ranged between $4.0x10^2$-$4.0x10^3$ cells/mL. Observed DOC values
ranged between 700-900 µgC/L and POC between 42-80 µgC/L and were within the range of expected values for the
oligotrophic Mediterranean (540—860 µgC/L for DOC and 9.6-104 µgC/L for POC) (Pujo-Pay et al., 2011). SSW TEP
concentrations ranged between $1.2x10^6$ and $1.1x10^7$ particles/L, with CSP between $5.6x10^6$ and $9.3x10^6$ particles/L, and will
be discussed in a future paper.
Enrichment factors (EF) in the SML relative to the SSW remained low with an average of 1.10 for DOC, 1.07 for bacteria,
and 1.17 for NCBL. As POC was not measured in the SML, we cannot report its EF. TEP was typically enriched relative to
the SSW, with an average EF of 4.5, while CSP EF was on average 2.7. Of importance, the dust deposition event that occurred
on June 4 lead to a drastic increase in SML dissolved iron relative to the SSW (EF ~800). This deposition event had important
impacts on the biology of the surface seawaters, which is the focus of another paper (Freney et al., 2020). As a result, TEP EF
increased to 17, bacteria EF increased to 1.5, and NBCL to 2.4. We next discuss the correlations between INP abundances and
biogeochemical properties in the following sections.

### 3.2.1 Correlations Between INP$_{SML}$ Abundance and Seawater Properties

**Table 1. Correlation statistics between INP$_{SML,-15C}$ and seawater properties in the SML and SSW, where p is the p-value test for significance and r is the Pearson correlation coefficient. Values in parentheses are calculated for days before the dust deposition event (i.e., days before June 4). Values that are not statistically significant (p > .05) are italicized.**

| Variable | p$_{all\ days}$ (p$_{pre\text{-}dust}$) | r$_{all\ days}$ (r$_{pre\text{-}dust}$) |
|---|---|---|
| **SSW** | | |
| CSP | 0.005 (*0.78*) | 0.87 (*-0.15*) |
| TOC$_{SSW}$ | 0.015 (*0.36*) | -0.85 (*-0.53*) |
| DOC$_{SSW}$ | 0.045 (*0.52*) | -0.76 (-0.39) |
| Nanoeukaryotes <10µm | 0.038 (*0.20*) | -0.63 (*-0.51*) |
| Micro-NCBL | *0.051* (0.021) | -0.70 (-0.88) |
| TEP | *0.25* (0.022) | *-0.46* (-0.88) |
| Bacteria HNA | *0.14* (0.043) | *0.57* (0.83) |
| **SML** | | |
| Dissolved Iron | .0000021 (.012) | 0.99 (0.91) |
| TEP EF | 0.00032 (*0.42*) | 0.95 (*0.41*) |
| Total Bacteria EF | 0.00075 (*0.82*) | 0.93 (*-0.12*) |
| CSP | 0.0053 (*0.25*) | 0.87 (*-0.56*) |
| Total NCBL | 0.0053 (*0.34*) | 0.87 (*0.48*) |
| Pico-NCBL | 0.0088 (*0.43*) | 0.84 (*0.40*) |
| Total Bacteria | 0.016 (*0.17*) | 0.81 (*0.64*) |
| Phytoplankton (NCBL+CBL) | 0.021 (*0.68*) | 0.78 (*-0.22*) |
| NCBL EF | 0.022 (*0.92*) | 0.78 (*0.054*) |
| DOC EF | 0.041 (*0.38*) | 0.78 (*-0.51*) |
| Nano-NCBL | 0.027 (*0.42*) | 0.77 (*0.41*) |
| Bacteria HNA | 0.012 (*0.068*) | 0.83 (0.78) |
| Bacteria LNA | 0.037 (*0.54*) | 0.74 (*0.32*) |
| TOC$_{SML}$ | *0.50* (0.020) | 0.31 (-0.93) |

Table 1 shows the correlation statistics between INP$_{SML,-15C}$ and selected observed seawater properties (SSW and SML), calculated either for all days of the PEACETIME experiment or only for days before the dust deposition event (i.e., days before June 4). Relationships are only listed in Table 1 if they were significant (p<.05) for either all days or pre-dust only days. Figure 4 shows the corresponding scatterplots of INP$_{SML,-15C}$ abundance and SSW properties. We note a statistically significant correlation between INP$_{SML,-15C}$ and CSP (r=0.87) as measured from the underway system. When considering only days before the dust deposition event, INP$_{SML,-15C}$ were significantly correlated with HNA bacteria (r=0.83) while the correlation with CSP is no longer statistically significant. INP$_{SML,-15C}$ are actually negatively correlated with most of the measured SSW properties either when excluding the dust event (for micro-NCBL$_{SSW}$ and TEP$_{SSW}$) or due to the dust event (for TOC$_{SSW}$, DOC$_{SSW}$ and nanoeukaryotes cell abundances). This points to a non-proportional transfer of each species from the bulk seawater to the SML relative to one another. Given the high p-values and weak correlation coefficients, it appears to be difficult to reliably relate INP$_{SML}$ to properties of the underlying SSW. Rather, we posit that INP in the SML are more reliably dictated by SML properties, as shown in the following paragraph.

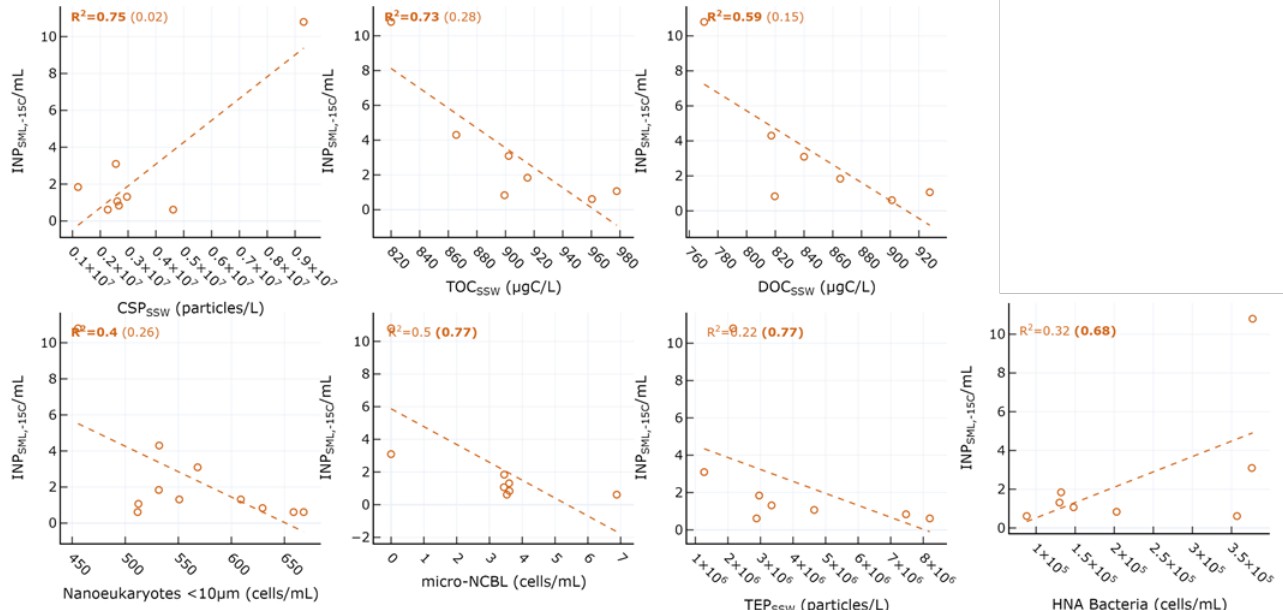

**Figure 4. Scatter plot of INP in the SML and various biogeochemical parameters in the SSW. $R^2$ for all days are shown in each plot, with $R^2$ calculated for only days before the dust event shown in parentheses. Statistically significant relationships are shown in bold.**

Figure 5 shows scatterplots of statistically significant relationships between $INP_{SML,-15C}$ concentrations and various SML properties. $INP_{SML,-15C}$ were most strongly positively correlated with dissolved iron (r=0.99), TEP EF (r=0.95), and bacteria EF (r=0.93). However, these relationships are skewed by the outlier due to the drastic increase in iron observed on June 4 (Figure S2a) from the dust deposition event, as described previously. It is difficult to discriminate between the dust and biological impact on the $INP_{SML,-15C}$, as dust is known to have good INP properties while also being capable of fertilizing the surface ocean with dissolved iron, leading to concomitant increases in biological activity. It is also possible that the dust deposition led to increased abundance of terrestrial OC, which would exhibit different INP activity. When considering days before the dust event, $INP_{SML,-15C}$ is only significantly correlated with dissolved iron (r=0.91) and TOC in the SML (r=-0.93). We note that while no longer statistically significant for pre-dust days, moderate correlations were still observed between $INP_{SML,-15C}$ and total NCBL (r=0.48), HNA bacteria (r=0.78), and total bacteria (r=0.64). Previous reports examining the correlation between INP and microbial abundance have yielded mixed results. For example, a report of INP in Arctic SML and SSW found no statistically significant relationship between the temperature at which 10% of droplets had frozen and bacteria or phytoplankton abundances in bulk SSW and SML samples (Irish et al., 2017). However, recent mesocosm studies using nutrient-enriched seawater found that INP abundances between -15°C and -25°C in the aerosol phase were positively correlated with aerosolized bacterial abundance (McCluskey et al., 2017).

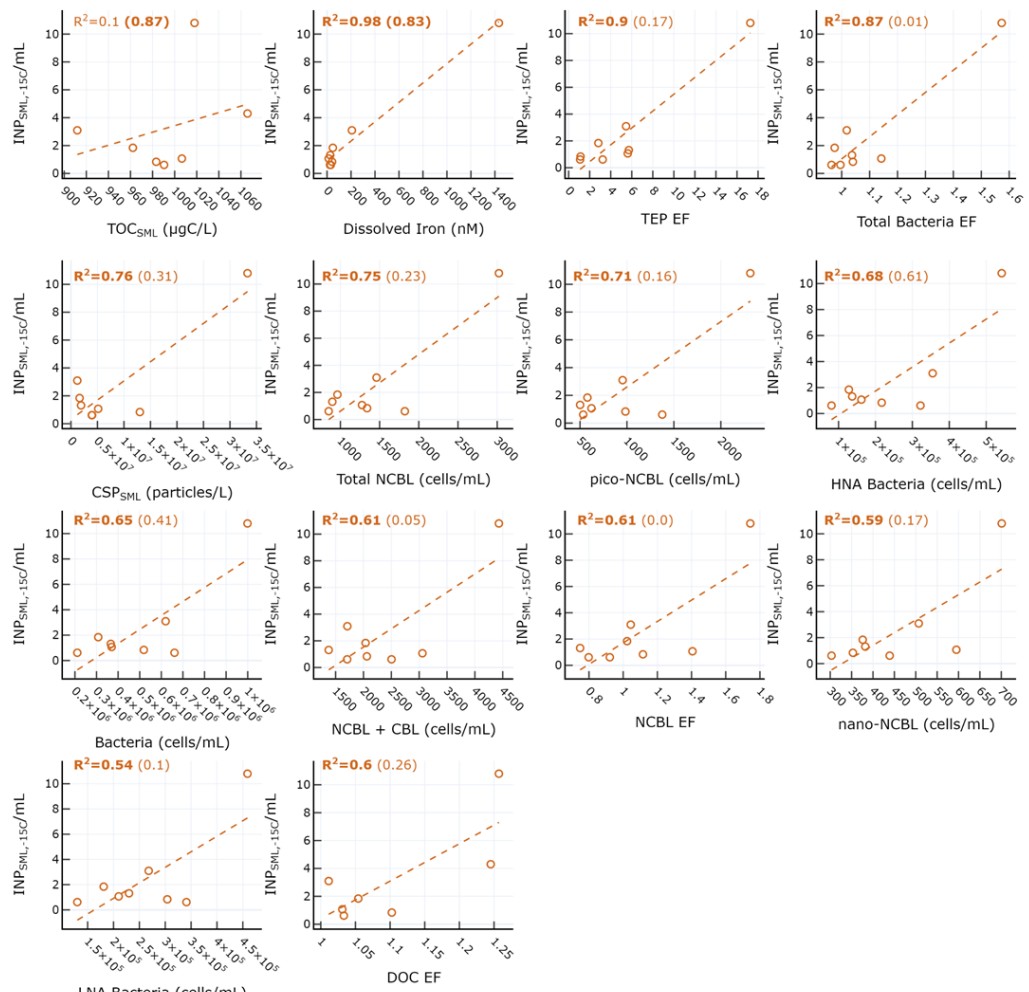

**Figure 5. Scatter plot of INP in the SML and various biogeochemical properties in the SML. $R^2$ for all days are shown in each plot, with $R^2$ calculated for only days before the dust event shown in parentheses. Statistically significant $R^2$ values are shown in bold.**

A previous study by Wilson and co-authors presented an INP parameterization (hereafter termed W15) based on a positive relationship between seawater TOC and INP abundance in Arctic, North Pacific, and Atlantic SML and SSW (Wilson et al., 2015). Total organic carbon in the SML ($TOC_{SML}$ µgC/L), derived here as the sum of POC in the SSW ($POC_{SSW}$) and DOC in the SML ($DOC_{SML}$), was poorly correlated with $INP_{SML,-15C}$ (r=0.31, p=0.50). Figure 7 shows the observed $INP_{SML,-15C}/TOC_{SML}$ ratio (INP per gram of TOC) for various temperatures and days of the experiment compared with the W15 parameterization (grey line). Our results show observed $INP_{SML}/TOC_{SML}$ ratios below those expected by the model proposed by W15, indicating the $TOC_{SML}$ in Mediterranean waters is less IN active at these temperatures than predicted by the W15 parameterization.

In agreement with our findings, a recent study found that the W15 model over-predicted observed INP concentrations in the aerosol phase during two separate mesocosm experiments (McCluskey et al., 2017) by assuming the INP/TOC ratio in the SML was preserved in the aerosol phase. The authors of that study speculated that the overprediction by the W15 model was due to the fact that it does not account for the complex transfer mechanism of organic matter from the SML to the aerosol phase. Our results here show that the overprediction by W15 persists even when calculating INP in the SML and therefore the overprediction may be due to other factors beyond the transfer of organic matter from the SML to the atmosphere. We stress however, that the TOC value used in this study was derived using $DOC_{SML}$ and $POC_{SSW}$ values as POC measurements in the SML were not conducted. As there typically exists an enrichment of organic matter in the SML relative to the bulk seawater, it is possible that the $POC_{SSW}$ we used to calculate $TOC_{SML}$ was below the actual POC content in the SML, thus underestimating $TOC_{SML}$. However, if this was the case, a higher abundance of $TOC_{SML}$ would only further increase the overprediction of W15

relative to our observations. Finally, it is possible that the oligotrophic nature of Mediterranean waters results in a pool of TOC with a different chemical composition than what is observed in more biologically productive waters such as the Arctic and Atlantic. For example, the pool of TOC during this study was dominated by DOC and featured low POC content, presumably due to low biological productivity.

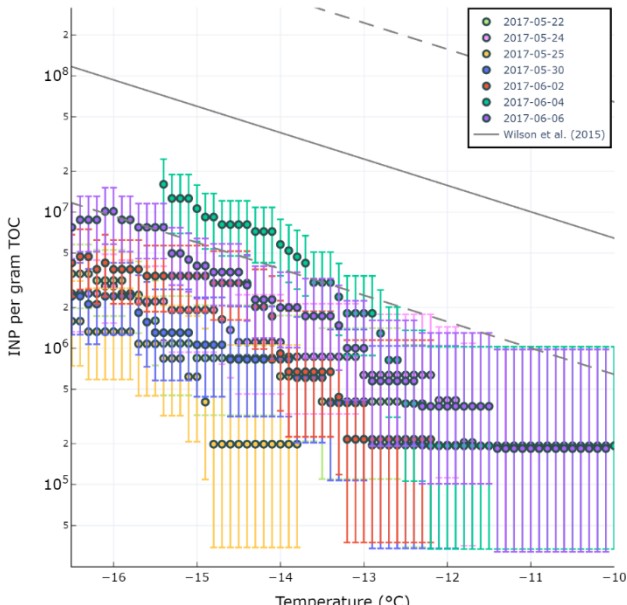

**Figure 6. Observed INP/TOC ratio in the SML during PEACETIME experiment for different temperatures. The gray line is the fit from Wilson et al., 2015.**

In summary, $INP_{SML,-15C}$ increased with SML microbial cell counts (e.g., NCBL and heterotrophic bacteria), $Fe_{SML}$ and $DOC_{EF}$ during a dust deposition event, but were overall not correlated with TOC nor DOC in the SML. Compared to previous studies, the INP/TOC ratio observed in the Mediterranean is low. We surmise that the overprediction of INP/TOC by the model may either be caused by a different relationship between INP and TOC at warmer temperatures, or possibly be due to the chemical characteristics of TOC in the oligotrophic Mediterranean. This complicated relationship between seawater TOC and $INP_{SML}$ highlights the need for further studies focused on the chemical composition of DOC and POC in bulk SSW and SML. Further experiments during low and high biological productivity are needed in controlled environments to better determine under what conditions (oligotrophic and eutrophic) and location in the water column (i.e., bulk SSW vs SML) TOC, bacteria, and phytoplankton are linked to INP across a range of temperatures. Finally, regardless of the exact mechanism, the impact of dust deposition on $INP_{SML,-15C}$ is fairly large, as we observe an increase of by $INP_{SML,-15C}$ by almost an order of magnitude during the dust event. This impact may have climate implications if $INP_{SML,-15C}$ were efficiently transferred to the sea spray.

### 3.2.2    Correlations Between $INP_{SSA}$ Abundance and Observed SSA and Seawater Conditions

In the following section, we compare $INP_{SSA}$ at various temperatures with seawater and SSA properties. Submicron particle concentrations ranged between 1000-3000 particles/cm$^3$ (Figure S3) and its dependence of seawater biology is further explored in a separate manuscript (Sellegri et al. under revision). For comparison with seawater properties, $INP_{SSA}$ was first normalized by SSA particle surface area ($0.1 < D_p < 10$ µm, Figure S4)(see methodology in Section 2.3.1 for estimation of SSA surface area for particles larger than $D_p=500$ nm).

Table 2 shows the correlation statistics between $INP_{SSA}$ normalized by SSA particle surface area and select conditions in the SML for relationships that were statistically significant. Figure 7 shows the corresponding scatter plots for these relationships. We also tested for correlations on days not affected by the dust event (i.e., days before June 4), and their statistics are shown in parentheses in Table 2 and Figure 7. Surprisingly, there were no significant correlations between $INP_{SSA,-18C}$ and

conditions in the SML, including TEP and CSP abundance and enrichment factors, bacteria abundance and enrichment factors, nor with $INP_{SML}$ as measured by the LINDA instrument. This is somewhat unexpected considering INP in the SML at -15°C was correlated with SML phytoplankton and bacteria counts, which are all expected to transfer efficiently from the SML to the aerosol phase, an assumption widely used in the modelling community. Similarly, -22°C $INP_{SSA}$ had no significant correlations with SML variables, except for TEP EF which was positively correlated (r=0.69) when only considering days before the dust deposition event. At -25°C, $INP_{SSA}$ were found to be significantly correlated with $DOC_{SML}$ and $TOC_{SML}$ on all days (r=0.82 and r=0.81 for DOC and TOC, respectively). When examining only pre-dust event days, the significant correlations included DOC enrichment as well as nano- and micro-CBL.

**Table 2. Correlation statistics between $INP_{SSA}$ and properties in the SML, where p is the p-value test for significance and r is the Pearson correlation coefficient. Values in parentheses are calculated for days before the dust deposition event (i.e., days before June 4). Values that are not statistically significant (p > .05) are italicized.**

| Variable | $p_{\text{all days}}$ ($p_{\text{pre-dust}}$) | $r_{\text{all days}}$ ($r_{\text{pre-dust}}$) |
|---|---|---|
| **-18˚C** | | |
| *No significant correlations* | | |
| **-22˚C** | | |
| TEP EF | *0.81* (0.026) | *-0.078* (0.69) |
| **-25˚C** | | |
| $DOC_{SML}$ | 0.0073 (0.00054) | 0.82 (0.94) |
| $TOC_{SML}$ | 0.017 (0.0066) | 0.81 (0.89) |
| DOC EF | *0.46* (0.014) | *0.28* (0.81) |
| Nano+Micro CBL | *0.10* (0.019) | *0.47* (0.72) |

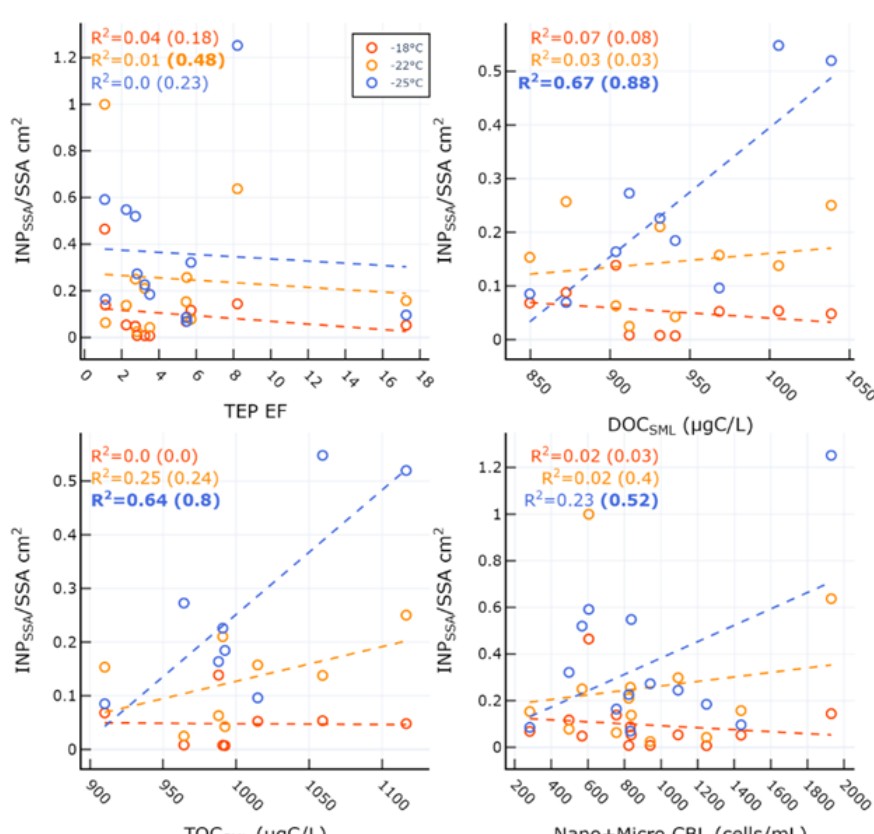

**Figure 7. Scatter plots of $INP_{SSA}$ normalized by SSA particle surface area at three temperatures and select conditions in the SML for relationships that were statistically significant. Corresponding correlation parameters are reported Table 2. $R^2$ values for all days are shown in each plot, with $R^2$ values for days not including the dust deposition event (i.e., days before June 4) in parentheses. $R^2$ for statistically significant relationships are shown in bold.**

**Table 3. Correlation statistics between INP$_{SSA}$ and properties in the SSW, where p is the p-value test for significance and r is the Pearson correlation coefficient. Values in parentheses are calculated for days before the dust deposition event (i.e., days before June 4). Values that are not statistically significant (p > .05) are italicized.**

| Variable | p$_{all\ days}$ (p$_{pre-dust}$) | r$_{all\ days}$ (r$_{pre-dust}$) |
|---|---|---|
| **-18°C** | | |
| POC$_{SSW}$ | *0.46* (0.0064) | *0.18* (0.63) |
| DOC$_{SSW}$ | *0.16* (0.020) | *-0.51* (-0.79) |
| **-22°C** | | |
| Nanoeukaryotes <10μm | 0.014 (0.050) | -0.53 (-0.48) |
| Prochlorococcus | *0.36* (0.000019) | *0.23* (0.89) |
| POC$_{SSW}$ | *0.29* (0.037) | *0.25* (0.54) |
| Coccolithophores | *0.72* (0.035) | *0.084* (0.51) |
| Micro-NCBL | *0.14* (0.0088) | *0.43* (0.77) |
| **-25°C** | | |
| Nanoeukaryotes <10μm | 0.0055 (0.0040) | -0.58 (-0.66) |
| Prochlorococcus | 0.00076 (0.00015) | 0.72 (0.84) |
| Coccolithophores | 0.031 (0.042) | 0.47 (0.50) |
| Cryptophytes | 0.028 (0.050) | 0.48 (0.48) |
| Micro-NCBL | 0.0012 (0.0049) | 0.79 (0.81) |
| Nano-NCBL | 0.048 (*0.058*) | 0.56 (*0.62*) |

Table 3 and the corresponding scatter plots in Figure 8 show that a weak correlation exists between INP$_{SSA}$ active at -18°C and POC$_{SSW}$ for all days, but becomes significant and strong for days not including the dust event. This points to the possible interference of a different class of organic carbon (e.g., terrestrial OC) or the introduction of some other IN active material (e.g., dissolved iron) which masks the impact of the original pool of POC$_{SSW}$ on INP concentrations. INP$_{SSA,-18C}$ are also significantly correlated INP$_{SSW,-16C}$, (results not shown) but with a sample size of n=4 this finding requires further validation. Nonetheless, this result could indicate that INP$_{SSA}$ at this temperature come from the bulk water rather than the SML. INP$_{SSA}$ at -22°C show a slightly weaker, yet still significant correlation with POC$_{SSW}$ than INP$_{SSA}$ at -18°C on pre-dust days (r=0.54). Additionally, they have a correlation with Prochlorococcus, coccolithophores, and micro-NCBL. This finding is in agreement with a recent study in which particles generated from lysed Prochlorococcus cultures exhibited good ice nucleating capabilities, albeit at much colder temperatures that observed in our study (i.e., T< -40°C) (Wolf et al., 2019). INP$_{SSA}$ at -25°C were correlated with similar variables as INP$_{SSA}$ at -22°C, with the exception POC$_{SSW}$. Furthermore, the correlations with the various microbial categories was stronger for INP$_{SSA}$ at -25°C than at warmer temperatures, indicating these parameters are more associated with cold INP. Interestingly, INP$_{SSA,-25C}$ was not correlated with DOC$_{SSW}$, yet was correlated with DOC$_{SML}$ (Table 2), potentially indicating an important step in the process of transfer of IN active DOC material to the atmosphere is its prior enrichment at the SML.

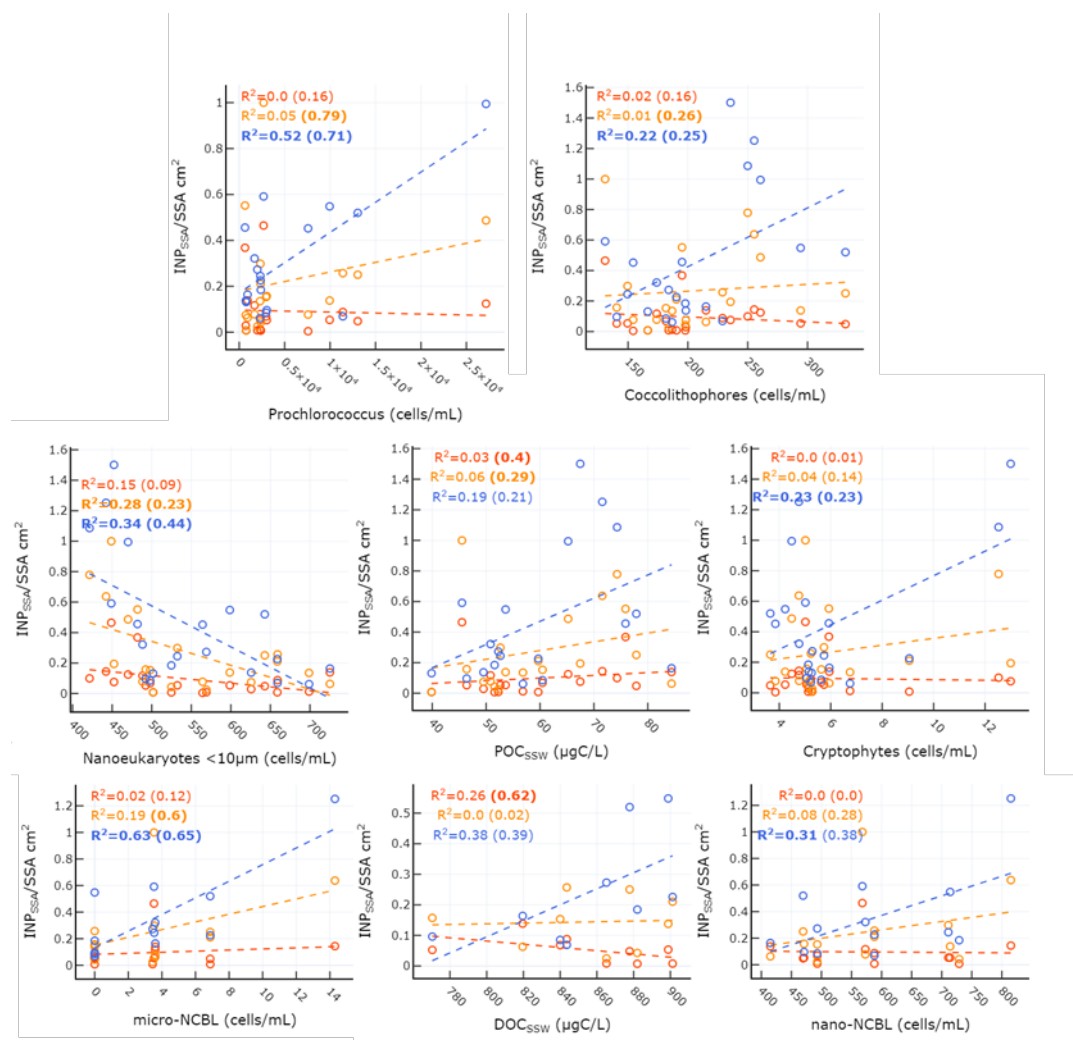

**Figure 8. Scatter plots of INP$_{SSA}$ normalized by SSA particle surface area at three temperatures and select conditions in the SSW for relationships that were statistically significant. Corresponding correlation parameters are reported Table 3. R$^2$ values for all days are shown in each plot, with R$^2$ values for days not including the dust deposition event (i.e., days before June 4) in parentheses. R$^2$ for statistically significant relationships are shown in bold.**

Table 4 and Figure 9 show the significant correlations between INP$_{SSA}$ and SSA properties. A positive correlation was observed between INP$_{SSA,-18C}$ and SSA organic carbon (OC) as well as the ratio of SSA water-soluble organic carbon to organic carbon (WSOC/OC). The correlation between WSOC/OC and INP$_{SSA,-18C}$ makes sense given the finding that INP$_{SSA,-18C}$ was correlated with POC$_{SSW}$. A higher WSOC/OC value would suggest a higher fraction of soluble organics which would be expected to transfer to the atmosphere from the bulk SSW rather than the SML due to their high solubility.

**Table 4. Correlation statistics between INP$_{SSA}$ and SSA properties, where p is the p-value test for significance and r is the Pearson correlation coefficient. Values in parentheses are calculated for days before the dust deposition event (i.e., days before June 4). Values that are not statistically significant (p > .05) are italicized.**

| Variable | p$_{all\ days}$ (p$_{pre-dust}$) | r$_{all\ days}$ (r$_{pre-dust}$) |
|---|---|---|
| **-18˚C** | | |
| WSOC/OC | 0.0099 (0.014) | 0.68 (0.68) |
| OC | 0.018 (0.021) | 0.64 (0.65) |
| WSOC | *0.25* (0.0074) | *0.29* (0.66) |
| **-22˚C** | | |
| WSOC | 0.042 (0.0082) | 0.48 (0.65) |
| OC | 0.015 (0.0080) | 0.66 (0.72) |
| WIOC | *0.061* (0.043) | *0.53* (0.59) |
| OMSS | *0.066* (0.028) | *0.52* (0.63) |
| **-25˚C** | | |
| WIOC | 0.037 (*0.057*) | 0.58 (*0.56*) |
| OMSS | 0.016 (0.025) | 0.65 (0.64) |



Figure 9 and Table 4 also show that $INP_{SSA,-25C}$ had a significant correlation with WIOC and organic mass fraction of sea spray
(OMSS) (r=0.58 and r=0.65, respectively). As mentioned above, $INP_{SSA,-25C}$ was found to be correlated with various microbes
in the SSW, specifically Prochlorococcus, coccolithophores, and nano- and micro-NCBL. Phytoplankton are known for their
ability to produce extracellular polymeric substances (Thornton, 2014), and a previous mesocosm experiment showed
microbially-derived long-chain fatty acids were efficiently ejected from the seawater as SSA, increasing the fraction of highly-
aliphatic, WIOC (Cochran et al., 2017). A separate manuscript discusses the trend and controls on SSA chemical composition,
linking the different classes of organic carbon in submicron SSA to seawater chemical and biological properties (Freney et al.,
2020). In this work, OMSS was linked to $POC_{SSW}$ and the coccolithophores cell abundance in the SSW. In light of this and
given the correlation of $INP_{SSA,-25C}$ with seawater microbial abundance and with SSA OMSS and WIOC, it seems likely that
$INP_{SSA}$ at this temperature are related to the exudates of phytoplankton which are concentrated at the SML and then emitted
into the SSA as WIOC.

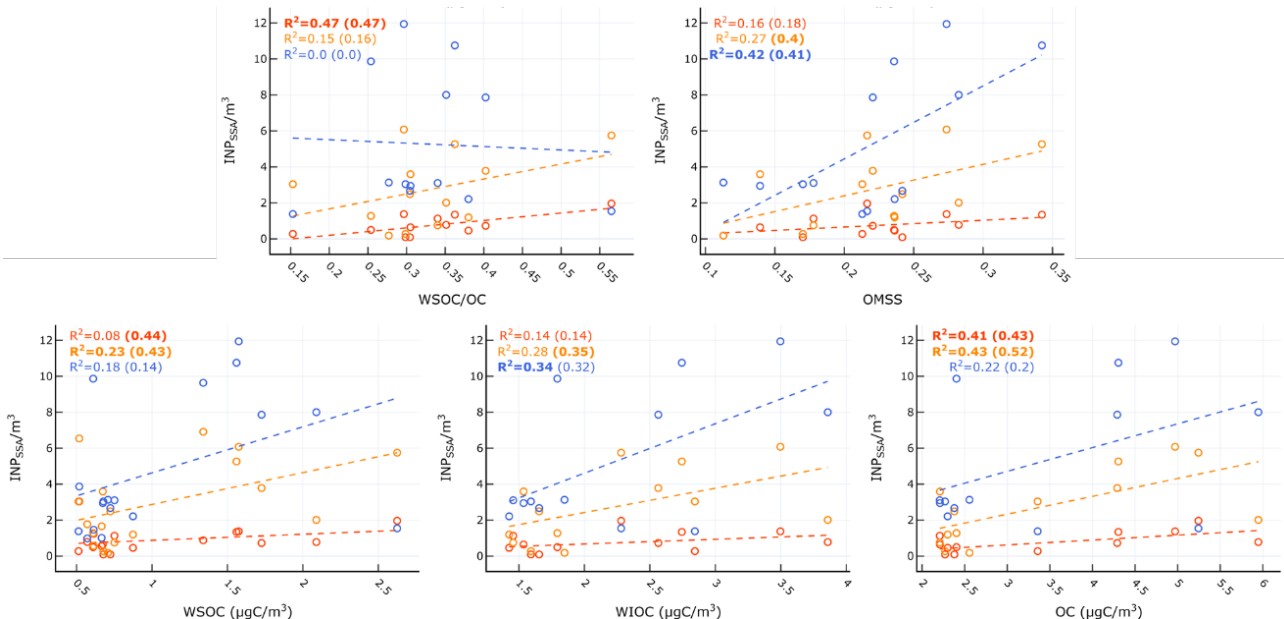

**Figure 9. Scatter plots of $INP_{SSA}$ at three temperatures and SSA properties for relationships that were statistically significant. Corresponding correlation statistics s are reported Table 2. $R^2$ values for all days are shown in each plot, with values calculated pre-dust event (i.e., days before June 4) in parentheses. Statistically significant values are shown in bold.**

To summarize the results thus far, we have found evidence for the existence of two classes of INP in SSA with
separate sources: 1) a class of INP related to POC in the bulk SSW and SSA WSOC and 2) a class of INP related to microbial
abundance and POC in the SSW, DOC in the SML, and WIOC in SSA. These findings of a two-component marine INP
population agree with a recent study which also reported on the existence of dual classes of INP emitted as SSA during two
mesocosm experiments, described as: 1) particulate organic carbon INP coming from intact cells or IN-active microbe
fragments and 2) dissolved organic carbon INP composed of IN-active molecules enhanced during periods when the SML is
enriched with exudates and cellular detritus (McCluskey et al., 2018b). However, in contrast to that study, we report here the
existence of separate temperature regimes at which each INP class is active. Here, the first class of INP consists of INP that
are more active at warmer temperatures (T=-18°C) while the second class of INP are active at colder temperatures (T=-25°C).
INP at T=-22°C correlates with items from both warm and cold categories.

## 4 Proposal of New INP Parameterization and Comparison with Previous Models

To date, parameterizations for the estimation of INP in SSA have not incorporated the knowledge of a two-component
INP population. Rather, they have predicted INP based on OC or SSA surface area (W15 and MC18, respectively). To improve

upon existing models, we formulated various parameterizations consisting of different time periods, features, and number of components for temperature ranges. Predictor features were chosen based upon their correlation with INP concentrations as described in the previous section. Single component parameterizations in which INP across all three temperatures were linked with the same features were compared with two-component parameterizations in which INP were split into warm and cold categories, each having their own predictor features. Finally, we developed and compared altered versions of the W15 and MC18 models to account for the oligotrophic seawater of the Mediterranean Sea, as the existing models were formulated from observations of eutrophic waters. Each parameterization was recalculated using data across all days of the cruise as well as for only days before the dust deposition event in order to determine the impact of the dust event on the ability to predict INP. The complete set of parameterizations and their associated fit metrics ($R^2$ and $R_{adj.}^2$) are given in Table S2.

Figure 10a shows observed vs predicted INP$_{SSA}$ for the W15 model, while Figure 10b shows the same but using the MC18 parameterization. Similar to our results for seawater INP (Figure 6), a large overprediction is found relative to our observations when using W15. Figure 10b shows that while MC18 is a slight improvement over the W15 approach, it still overpredicts INP by two orders of magnitude. We also present re-calculated best-fit-lines to data using the same features as in W15 and MC18 (i.e., OC and SSA surface area) in order to account for possible changes due to the oligotrophic nature of the Mediterranean Sea. We term these two parameterizations the altered Wilson fit for oligotrophy, which is given by:

$$\frac{INP}{m^3} = \exp\left(-7.332 - (0.2989 * T) + (0.3792 * OC_{SSA})\right)$$

and the altered McCluskey fit for oligotrophy, given as:

$$\frac{INP}{\mu m^2} = \exp\left(-26.57 - (0.2782 * T)\right)$$

The results for these fits are shown in Figure 10a,b alongside the results of the original W15 and MC18 parameterizations. Both altered models offer improvements over the original parameterizations. The adjusted $R^2$ of the altered Wilson fit for oligotrophy on log-transformed INP abundance was $R_{adj}^2=0.59$ and was $R_{adj}^2=0.32$ for the altered McCluskey fit for oligotrophy. Interestingly, the adjusted Wilson fit for oligotrophy performs better than the adjust McCluskey fit for oligotrophy, which is the opposite of what was found when comparing the original models.

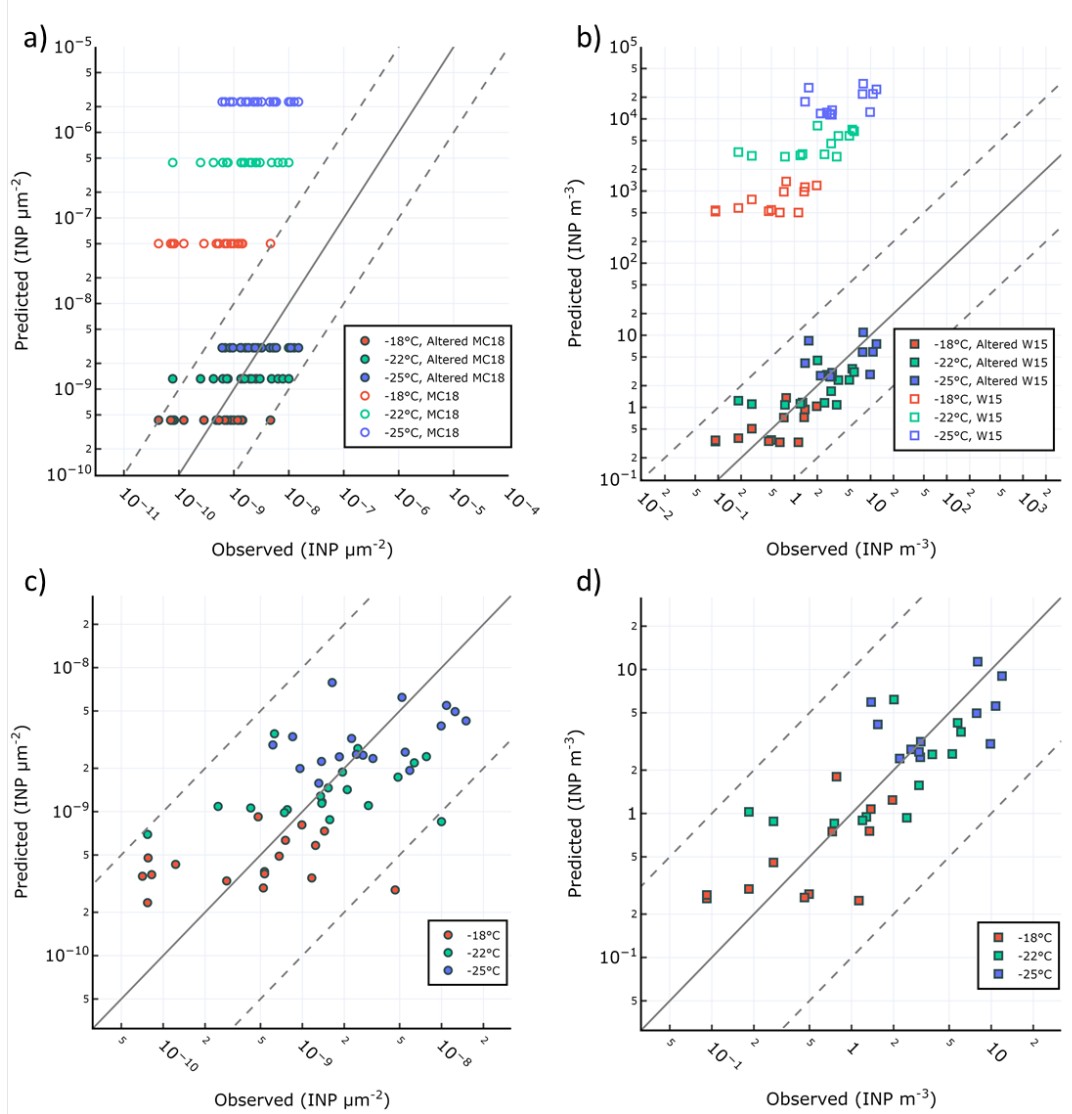

**Figure 10. Different parameterizations for prediction of INP in SSA. a) W15 and refit of same method using PEACETIME observations b) MK18 and refit of same method using PEACETIME observations c) single-component parameterization for INP/μm² SSA surface area where INP at all temperatures are related to POC_SSW d) two-component parameterization for INP/m³ where INP≥-22°C are related to OC and INP <-22°C are related to WIOC.**

We also tried a range of novel parameterizations based on the observed correlations between INP_SSA with seawater and SSA properties. Below we describe two parameterizations which offered good fits to the data. The single-component parameterization assumes the abundance of INP per unit surface area of total SSA at each temperature can be predicted from POC_SSW concentrations:

$$\frac{INP}{\mu m^2} = \exp(-28.5324 - (0.2729 * T) + (0.0361 * POC_{SSW}))$$

The second parameterization separates INP into warm and cold classes, where warm INP (≥-22°C) are related to SSA OC and cold INP (<-22°C) are related to the concentration of SSA WIOC. This two-component parameterization predicts the concentration of INP/m³ through the following equations:

$$\frac{INP_{T \geq -22°C}}{m^3} = \exp(-7.9857 - (0.3178 * T) + (0.4643 * OC_{SSA}))$$

$$\frac{INP_{T < -22°C}}{m^3} = \exp(-6.6606 - (0.2712 * T) + (0.5755 * WIOC_{SSA}))$$

Figure 10c,d shows the results of our single-component model using POC_SSW and the two-part model which uses SSA WIOC and OC and considers the separate temperature classes of INP. The adjusted $R^2$ for each model on the log-transformed INP abundance were $R_{adj}^2$=0.404 for the single component model using POC_SSW and $R_{adj}^2$=0.60 for the two-component model using OC and WIOC. This result reveals that they both fit the observations better than the altered McCluskey parameterization

for oligotrophy, while the two-component method performs as well as the altered Wilson parameterization. Each parameterization's fit to the data is improved when considering pre-dust days only ($R_{adj}^2$=0.63 for the two-component parameterization and $R_{adj}^2$=0.57 for the single-component parameterization). The improvement is more pronounced for the single-component parameterization using $POC_{SSW}$, further pointing to the fact that such dust deposition events can alter the INP properties of surface waters and the subsequent SSA, either through introduction of terrestrial OC or by triggering changes to the trophic status of the surface waters, resulting in a different class of biologically produced OC. We note that the ratio of $INP_{SSA,-18C}/OC_{SSA}$ is on average $2.08 \times 10^5 \pm 1.4 \times 10^5$ INP/gC while the ratio of $INP_{SML,-15C}/TOC_{SML}$ as reported in Section 3.2.1 is $3.2 \times 10^6 \pm 3.5 \times 10^6$ INP/gC. This points to a depletion in the abundance of INP active material by a factor 16 as it transfers from the seawater to the SSA, which is typically assumed to be negligible in modelling studies. However, when available, using a ratio of $INP_{SSW}/TOC_{SSW}$ to predict sea spray originating INP in the atmosphere seems a better approach than using the ratio $INP_{SSW}/NaCl_{SSW}$. Finally, we remind the readers that the two-component parameterization uses results of SSA chemistry for submicron particles only. As previous studies have shown that the overwhelming majority of SSA OC is found in the submicron phase (Gantt and Meskhidze, 2013), we argue that our analysis of WIOC, WSOC, and OC concentrations in submicron SSA is representative of the whole size range of SSA.

**5 Conclusions**

In this paper we have presented results from the month-long PEACETIME cruise which took place in the Mediterranean Sea during the spring of 2017, which was characterized with a dust wet deposition event that occurred towards the end of the cruise. First, we find that the INP concentrations measured in the seawater are in agreement with previous studies on oligotrophic waters (Gong et al., 2020). Observed INP $cm^2$ of SSA are below those reported in the literature, likely due to differences in biological activity of source waters. We next investigated the relationship between seawater INP concentrations and seawater biogeochemical properties. In the SML, the increase of $INP_{SML,-15C}$ concentrations during the dust deposition event followed the SML microbial cell counts (e.g., NCBL, CBL and heterotrophic bacteria), $Fe_{SML}$ and $DOC_{EF}$. Excluding this dust event, $INP_{SML,-15C}$ were still correlated to Fe and bacteria (although not significantly) in the SML. Overall $INP_{SML,-15C}$ were not correlated with TOC nor DOC in the SML and compared to previous studies, the INP/TOC in the SML observed during the PEACETIME cruise was low. We surmise that these low INP/TOC is a result of TOC from the oligotrophic Mediterranean being less IN active.

The impact of dust deposition on $INP_{SML,-15C}$ is fairly large, as we observe an increase of $INP_{SML,-15C}$ by almost an order of magnitude during this event. This impact of dust deposition could have climate implications if $INP_{SML,-15C}$ were efficiently transferred to the sea spray emitted to the atmosphere. However, we find that $INP_{SSA}$ does not evolve in the same manner as the $INP_{SML}$ does, as an increase of $INP_{SSA}$ is observed with at least a three day delay after the dust wet deposition event. This difference could be attributed to the fact that $INP_{SSA}$ measured at -18°C are more influenced by the INP concentration in the bulk surface seawater (as shown by the correlation between $INP_{SSA,-18C}$ and $INP_{SSW,-16C}$). It is possible that IN active species deposited during the rain event, either dust- related or biology-related, take a few days before entering the bulk surface layer.

We also investigated the relationship between $INP_{SSA}$ and various biogeochemical values in the SML, SSW, and SSA. In general, we observed the existence of two classes of $INP_{SSA}$, each linked to different classes of organic matter. Our results indicate each class is active at separate temperatures. Warm INP ($INP_{SSA,-18C}$) are linked to water soluble organic matter in the SSA, but also to SSW parameters ($POC_{SSW}$ $INP_{SSW,-16C}$). This indicates that INP at this temperature come from the bulk water rather than the SML. Colder INP ($INP_{SSA,-25C}$) are rather correlated with SSA water-insoluble organic carbon, and SML properties (DOC). As the cold INP are also correlated to the SSW nano- and micro-NCBL cell abundance as well, we hypothesize that these classes of phytoplankton produce surface-active water-insoluble organic matter that is active as IN at these temperatures and are transferred to the atmosphere via the SML. Unfortunately, we do not have measurements of the "colder" temperatures INP in the SML to check this hypothesis.

We finally proposed a single-component model linking INP/m$^3$ to POC$_{ssw}$ and a two-component model linking warm INP to SSA OC and cold INP to SSA WIOC. Both models utilize features that are readily approximated either from satellite data, biogeochemical models, or from existing parameterizations and observations (Aumont et al., 2015; Rasse et al., 2017; Albert et al., 2010). We then showed these parameterizations fit the data much better than previous single component models based solely on SSA surface area (MC18) or OC content (W15), developed from studies of more biologically active waters. We also re-calculated parameterizations based on SSA surface area and SSA OC content but for the oligotrophic Mediterranean Sea. The parameterization using SSA OC content fits almost as well as the two-component model using SSA OC and WIOC. However, given the results of correlation analysis with SSA properties as well as results from previous studies indicating a dual composition of INP, we believe the two-component model should help improve attempts to incorporate marine INP emissions into numerical models.

**Acknowledgements** This study is a contribution to the PEACETIME project (http://peacetime-project.org), a joint initiative of the MERMEX and ChArMEx components supported by CNRS-INSU, IFREMER, CEA, and Météo-France as part of the programme MISTRALS coordinated by INSU. PEACETIME was endorsed as a process study by GEOTRACES. PEACETIME cruise https://doi.org/10.17600/17000300. We thank the captain and the crew of the R/V Pourquoi Pas? for their professionalism and their work at sea. The underway optical instrumentation was provided by Emmanuel Boss's group funded by Nasa Ocean Biology and biogeochemistry. This work has also received funding from the European Research Council (ERC) under the European Union's Horizon 2020 research and innovation program (Sea2Cloud grant agreement No 771369). Sea2Cloud was endorsed by SOLAS.

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
