# Peer review of "Particles Based on Seawater Biology and Sea Spray Aerosol Measurements in the Mediterranean Sea"

_Atmospheric Chemistry and Physics, 2020_

## Referee Comment (RC1) · Anonymous Referee #1 · 30 Jul 2020

ACP Review of 'A Two-Component Parameterization of Marine Ice Nucleating 2 Particles Based on Seawater Biology and Sea Spray Aerosol 3 Measurements in the Mediterranean Sea'. *please see PDF for clear technical corrections.

General Comments: The paper Trueblood et al. 2020 is a nice study which considers INP data from oligotrophic/Mediterranean waters and shows that eutrophic parameterisations (W15 and MC18) result in over-prediction. The occurrence of a dust deposition event over the measurement periods, in conjunction with measurements from the SML, SSW, and SSA, makes for very interesting reading, although it is a shame that

the dataset ends before INPSSA concentrations reached a clear maximum. However, this brings up the question can the two-component temperature dependent parameterisation from this study be relevant to much larger bodies of water? I am happy that the authors themselves addressed this in the need for future work relating INPSSA to POC and NCBL measurements in the Southern Ocean. However, the difficulty in choosing POC and NCBL in relationship to INP is that all variables must be directly measured. The authors give no indication how to apply this parameterisation in a global model. The largest problem with the current study is that no uncertainties or error in the INP measurements (or biological measurements for that matter) are shown or discussed. This paper should not be published without the addition or evaluation of the inherent errors and uncertainties in the measurements themselves and the application of the measurements to creating a parameterisation. Also, the authors have not convincingly shown that the temperature dependent parameterisations are necessary to model INP concentrations, although they have shown that oligotrophic waters may need different parameterisations to eutrophic waters. A question also arises of whether INPSSA increases after the dust event are really to do with the dust event or not? INPSSA did not seem to be very connected with SML conditions (which surprised the authors and may therefore necessitate more attention). Lastly, throughout the text, Figure and Table descriptions are kept too short and often do not fully describe what is shown.

Specific Comments: Temperature nomenclature (TM) varies throughout the text, sometimes for example as -15C or -15 C. Please keep consistency and it is suggested to use the proper format of e.g. -15°C. All figures appear blurry, this should be corrected. Line 102 – SSW properties were obtained from two depths 20 cm and 5 m, why this is done at two depths is never explained. It is important as POC is measured at 5 m depth while SML and 20 m depth SSW samples were measured simultaneously and both calculated NCBL. Line 110 – Why is there a specific empirical relationship for PEACETIME? Will this affect other estimations of POC used for the parameterization? Line 126 – methodology should be described in brief, or else simply cited if it is the only established measurement practice. Line 140 – calculation should be described in

brief. What are the associated errors/uncertainties of this methodology using LINDA? Line 153 – You talk about bin size or 100-500 nm, but what is this? Is it the dry particle (electrical?) mobility diameter? This must be states explicitly. Line 161-164. Confusing description of how measurement of WSOC was measured vs how TOC was measured. Then how was WIOC measured? Line 166-175. It seems that no measurements of ambient INP were taken. This seems concerning as often tank and ambient measurements do not always compare well to one another. Do you have evidence that the plunging jet SSA measurements were similar to that of the ambient SSA over the Mediterranean? Line 166-175. What are the associated errors/uncertainties of this methodology using DFPC? Line 178 – 183. How are INP from SSW measured (I assume it is LINDA – but this is not included in your methodology)? Which SSW measurement is tested for INP? Line 191 – 192. The use of the term 'peak' here is a bit confusing in two ways. Purely graphically it is true that INPSSA,-25C peaked on May 12, however, the implication that it is truly peaking is false as this is the first measurement it could have been higher before measurements commenced. Has contamination of the plunger tank system been ruled out as it is by far the greatest disparity between different temperatures for INPSSA? Line 196. Again the use of the word peak is a bit misleading as measurements ended before the true peak could be observed. In this case can you really comment on the time difference between one peak in SML and SSA? Line 204. Were there any differences in cell counts between SSW at 5 m or 20 cm depth? Line 206-207. Are these the ranges associated with Pujo-Pay et al. 2011, or the ranges for this study? If the latter than perhaps give the expected range as well. Line 209. How are you calculating enrichment factor? It is good to state as sometimes confusion arises. Line 236-240. This paragraph feels like it is out of place as a discussion paragraph crammed between the synopsis of the results in the same Figure. It does not add much to the discussion. What do these two studies mean for your results? If anything they imply that you must compare INPSSA to SSA bacterial abundance. Line 250. DOC EF is positively correlated with INPSML,-15C, and you state this is due to the dust event, and in the next statement say that the fraction of

DOC enriched in the SML during the dust event has specific IN properties. It seems possible that the DOC came from non-marine originating bacteria and that the deposition event also deposited terrestrial DOC which is the origin of increased IN ability. Or more so, could the correlation be coincidental with another correlating factors from the dust event (i.e. Fe)? No indication is given of why the authors believe it to be 'likely connected to the CSP abundance, albeit not to the TEP', which if given may add value to the statement. Line 291-293. What are the total particle counts referred to in Line 292? How are they measured and how do they match well with SSA counts in the range? In terms of SSA surface area: (1) how was SSA calculated from Dp? (2) SSA have two noticeable modes larger than 500 nm, one is a submicron mode and the other is the jet-drop mode which are found to have mean dry mobility diameters near at 0.83 (Ovadnevaite et al. 2014) and $\sim$2 $\mu$m (Wang et al. 2017, Lewis & Schwartz 2004), respectively. According to Figure S3, most of the surface area distributions have already peaked by 0.5 $\mu$m particle diameter (with the possible exception of 2017-05-17), yet a significant portion of surface area for particles with Dp> 0.5 $\mu$m seems to be lost. It seems an overstatement to say 'most of the surface area of sea spray is comprised between this size range'. Ovadnevaite, J., Manders, A., de Leeuw, G., Ceburnis, D., Monahan, C., Partanen, A. I., Korhonen, H., and O'Dowd, C. D.: A sea spray aerosol flux parameterization encapsulating wave state, Atmos. Chem. Phys., 14, 1837-1852, 10.5194/acp-14-1837-2014, 2014. Wang, X., Deane, G. B., Moore, K. A., Ryder, O. S., Stokes, M. D., Beall, C. M., Collins, D. B., Santander, M. V., Burrows, S. M., Sultana, C. M., and Prather, K. A.: The role of jet and film drops in controlling the mixing state of submicron sea spray aerosol particles, Proceedings of the National Academy of Sciences, 114, 6978-6983, 10.1073/pnas.1702420114, 2017. Lewis, E. R., and Schwartz, S. E.: Sea Salt Aerosol Production: Mechanisms, Methods, Measurements and Models-A Critical Review, American Geophysical Union, 2004. Line 313. What is the difference between SSA OC and TOC here? How is OC calculated from the SSA? Line 326/327. It would be good to state the relevant conclusions of Freeney et al. 2020. Line 368. How are you calculating OMSS? Why is this in agreement with Cochran et

al. 2017? Line 413. It is stated that '…the INP concentrations measured in the SSW are in line with the INP measured in the SML…'. There is only one comparison of INP shown of the two (figure 2a) and only one temperature is shown for the SSW. Is there further evidence to back this statement? Indicate what evidence is referred to in the text. Table 1 – Description of table needs to state what p, R(R2) and n are. Is the p value of NCBL EF 0.78? This looks like a typo. Review the rest of the table to double check for other typographical issues. Why does it say CSPabundance, when in there is no explanation of the difference between CSP and CSPabundance? Table 2. Description of table needs to state what p, R(R2) and n are. The table is stretched over a page break. This should be corrected to be on one page. Change POC to POCSSW. Table 3. Description of table needs to state what p, R(R2) and n are. Figure 1. The image is blurry. The points indicated on the map are names with abbreviations that are never explained nor referred to in the text. If these refer to the dates mentioned in other graphs, this should be made clear. If not, then why are they there? Figure 2. Why is there no uncertainty associated with each measurement? INP measurements have some of the largest uncertainties in aerosol science, this can't be neglected. How do you explain why INPSSA,-25C and INPSSA,-22C are sometimes anti-correlated and sometimes not? Some other minor corrections are needed. This graph is blurry and should be higher resolution. It would be nice to have different keys for a) and b). The y-axis in a) should be written scientifically – i.e. either 10,000 or 1x104. It is difficult to differentiate the colours, effort should be taken to use different markers. The bottom access should probably be the 'Date' not 'Day Number' (see same issue in other graphs). Figure 3. This figure is also blurry with no error/uncertainty on the measurements shown. Figure 4. Y-axis scale is difficult to interpret, should be written for example 108 not 108. On the x-axis the authors might consider writing Temperature (°C) rather than (C). Again error/uncertainties should be shown, or else noted that the error bars are not larger than the data points. The description of Figure 4 is on a different page than the figure, this should be corrected. It is difficult to tell day=2017-05-24 from day=2017-06-06. The authors could probably omit the 'day=' in the key and make the text larger. Figure

5. Description does not mention INP normalised to SSA. Why use /cm3 rather than /nm2 which is what the surface area is shown in in Figure S3? When you normalise INP to SSA, should it not still be in term of (/m3 of air Âů SSA cm2)? Top left panel, should read '3x10-4' not '3x10-4'. Figure 6. Description should be below figure, and should include some more details of the graph. The figure is blurry, and need to be corrected. OMSS not explained. Figure 7. Description should mention only significant correlations shown. Text should not state that these panels are a matrix. The scatter plots are blurry and should be corrected to higher resolution. The authors may choose to add r-values to each panel to make it easier for readers to study the results. Figure 8. Graph should be made larger and enhanced to be less blurry. Y-axis scale is difficult to interpret, should be written for example 101 not 101. It is difficult to read the axes. Your 3 panel axes seem to be in different units, some per L and some per m3. These are all SSA INP so they should be terms of their atmospheric concentration. This should be explained in the description. Additionally, it seems clear from the graphs that while both the W15 and MC18 models over predict INP concentrations the over prediction is not really temperature dependent. The graph seems to show more of the difference between oligotrophic waters and eutrophic waters. How much does the authors' own parameterisations differ if only the colder (eq. 2) or warmer (eq. 1) parameterisation is applied to all the results? Are there any data of eutrophic waters which suggest a temperature dependence might improve the agreement? Supplementary Info – consider adding a schematic of measurements taken from the tank. Table S1. Usually tables come before Figures. Description of table needs to state what p, R(R2) and n are. Place a '0' before all values in column p. Figure S1. Where possible, missing data should be deleted rather than shown as a line jumping from the last measured point to the next. There should be graph panel specific keys as each factor is not shown on every graph. It would be nice if more detail could be given in the description of where/how these measurements were taken. A description of what POC or biovolume covers here could also be useful. Figure S2. Grey outline squares around a) and b) are somewhat off centre and cut-off the a) and b). Fe axis should be shown on the same scale in a)

and b). It would be nice to see INPSSA measurement overlaid in time with those SML and SSW conditions considered to be contributing most prominently to INPSSA concentrations. Figure S3. It is nearly impossible to tell some of these 'variable' apart as the same color is used for multiple days. Please graph in such a way that the surface area spectrums can be identified for each variable. If they are daily averages than the stdev should also be graphed. Y-axis, change from '(nmˆ2/(cmˆ3))' to '(nm2/cm3)'. The authors could probably omit the 'variable=' in the key. Also, it is low resolution. What is a scanotron? Were these not measured by the DMPS as stated in the methodology? Figure S4. Color of 'variables' again overlap for multiple days. Please graph in such a way that the number size distribution spectrums can be identified for each variable. If they are daily averages than the stdev should also be graphed. The y-axis shows dN/dlogDp in '(particles/(cmˆ3 nm))' the extra nm is likely a typo? It should be '(/cm3)'. The authors could probably omit the 'variable=' in the key. Also, the graph resolution is low. What is a scanotron? Were these not measured by the DMPS as stated in the methodology?

Technical corrections: Line 22 – delete the 's' after INP, as INP is defined plural earlier. Line 29 - delete the 's' after INP. This occurs many more times so check throughout the text. Line 33 – delete extra space '...to SSW parameters (POCSSW...'. Add an 'and' or a ';' between '(POCSSW INPSSW,-16C)'. Line 56 – delete ')(' between references and replace with ';'. Delete '-' after SSA. Line 62/63 – refer to study simply as 'Wilson et al. 2015 identified a temperature-dependent...'. Either delete the 's' from the end of the word entities or from concentrations. Line 68 – TM (see specific comments). Line 85 - delete the 's' after INP. Delete the 'the' before title of study. Here is it the title of the cruise or study? I suggest replace the word 'cruise' with 'study' and delete 'study' from the end. Line 87 – add space, 'May 10 – June 10, 2017'. Line 88 – delete 'were'. Line 92 – what is 'R/V'? Here 'Pourquoi Pas?' is written differently than later. Keep consistency. Line 94 – replace 'fashion from 35° to 42°' to 'fashion between 35° to 42°'. Line 109 – HPLC acronym not explained. Line 113 – FWS and SWS acronym not needed as never used again. Line 124 – ICP-MS acronym not

explained. Replace with full title as acronym not needed. Line 130 – MQ acronym not explained. Replace with full title as acronym not needed. Line 131 – add space between '0.5L' Line 132 – HCL acronym not explained, although it is well known as Hydrochloric acid. Authors may choose to spell it out as it is not repeated. Line 135 - add space, 'May 22 - June 7'. Line 137 – TM (see specific comments). Line 146 – change meter to 'm' Line 147 – ACSM acronym not described. DMPS and CPC acronym used before description. Line 153 – correct to '10-500 nm'. Line 159 – MSA acronym not explained. Replace with full title as acronym not needed. Line 160 – KOH acronym not explained. Replace with full title as acronym not needed. Line 161 – WSOC acronym used for first time and is not defined. Line 166 – '24h' change to '24-hour' to keep consistency. Line 167 – delete the 's' after INP, as INP is defined plural earlier. Line 169 – change to '47 mm' with space. Line 173 – TM (see specific comments). Add '. . .(for air temperatures of . . . -22.3 C, respectively)'. Line 175 – add 'INP/volume of air' Line 181/182 – TM (see specific comments). June 4 not 4th. Line 192 – use scientific notation for INP/m3 (i.e. 1.47x10-2 not 14.7x10-3). Line 196 – the peak in INPSSA occurred three days after INPSML peaked, not one day. Unless the authors meant to suggest that INPSSA only saw an increase begin a full day after the INPSML peak? Line 200 – Delete '(SI)'. Line 204/205 – keep same scientific notation for describing cells/mL. Line 209 – add 'Enrichment factors (EF). . .' Line 214 – delete 'next' (optional). Line 220 – consider adding in '. . .positive or negative correlations. . .' Line 254 – uppercase L for litre, such that 'TOC $\mu$gC/L'. Replace 'particulate organic carbon' with POC. Line 255 – Replace 'dissolved organic carbon' with DOC. Line 256 – Should be '(INP per gram of TOC)' not 'OC'. Is this cumulative INP as in W15, or is this INP/mL? Line 282 – Do you mean '. . .between seawater OC' or 'TOC'? Line 291 – add space between '500' and 'nm'. Line 294 – Only normalised size distribution shown in Figure S4, not number concentration. Perhaps add it in the graph key? Replace 'dependence of' with 'dependence on'. Line 298 – add space between '500' and 'nm'. Line 300 – add 'in' ahead of 'Table 2'. Line 307 – Give correlation stats for INPSSW,-16C Line 351 – replace 'the' with 'that'. Line 353 – Replace 'At this

temperature, INPSSA' with just 'INPSSA,-25C...' Line 361 – some overlap issue with graph and line numbering. Line 362 – change 'R=.84' to 'R=0.84'. Check for other numbering mistakes throughout the text. Line 380/381 – TM (see specific comments). Line 392 & equations – Warm INP defined as $\geq$-24C, but in eq. (1) says -22. Also, in eq. (1) 'POC' should be rewritten 'POCSSW' to keep clarity (unless authors want any POC to be used in which case more explanation should be given). Line 393 – this entire line should come before eq. (1) and (2). Line 425 –INPSWL? Change 'INPSWL and INPSML' to 'INPSSW and INPSML'. Line 430 – Is INPSSA measured at -16C or it -18C? Leave and 'and' or ',' between POC and INP. Line 436 – '...seawater POC and SSW microbial abundance' seems redundant or repetitive. Line 446 – it is written here 'RV' but elsewhere 'R/V'. 'Pourquoi Pas ?' is also written differently elsewhere.

Please also note the supplement to this comment:
https://www.atmos-chem-phys-discuss.net/acp-2020-487/acp-2020-487-RC1-supplement.pdf

---

## Referee Comment (RC2) · Anonymous Referee #2 · 18 Aug 2020

Review of "A Two-Component Parameterization of Marine Ice Nucleating Particles Based on Seawater Biology and Sea Spray Aerosol Measurements in the Mediterranean Sea" by Trueblood et al.

General comment

This study investigated the ice-nucleating abilities of surface microlayer (SML), surface seawaters (SSW), and sea spray aerosol (SSA) particles collected/generated in the Mediterranean Sea. In parallel to the evaluation of the ice-nucleating abilities of

the different samples, a large set of biogeochemical analysis were performed on the samples to understand the relationship between ocean biology and marine ice nucleating particles (INP). While the ice nucleation analysis of the SML and SSW samples was performed with a LED based Ice Nuclei Detection Apparatus (LINDA), the analysis for the SSA samples was performed using a Dynamic Filter Processing Chamber (DFPC). Taking into account the collected information, the authors developed a new two-component parametrization. Although the information collected/derived by the authors is very rich and valuable, in addition that they were collected in a poorly explored region on Earth, the manuscript is not easy to follows, it lacks important information, and the conclusions are not clearly supported by the provided data. The manuscript fits with the ACP scope, but the current version cannot be accepted for its publication.

Major Comment 1. Two different techniques were used to analyze the ice nucleating abilities of the samples, i.e., the LINDA and the DFPC. While the LINDA determines the INP concentrations via the immersion freezing mode, the DFPC does it via condensation freezing. Given that both data sets were used to develop the parametrization, I am wondering if the INP concentration delivered by both instruments are directly comparable. For example, I am wondering about the very low concentration of INPs reported for the SSA samples in comparison to literature data. It is a true number or is it an artifact related to the used method? 2. I was unable to fully understand how the INP concentrations for the SSA samples were obtained as the DFPC was not properly described. Is this a new custom-made instrument? Is this the first data delivered/published by this instrument? If this is the case, a much deeper description needs to be provided. If this is not the case, how good is the agreement of the data delivered by the DFPC against other well-known ice nucleation instruments? 3. I am surprised that the INPs concentrations of the SML, SSW, and SSA are not compared to literature data. Actually, a recent study by Gong et al. (2020) who also studied the SML, SSW, and SSA is not cited/discussed here. There are also other studies in marine environment performed at subtropical and tropical latitudes that deserved to be discussed in the context of the present study. 4. Given that the chemical composition of the SSA is linked to the size

of the aerosol particles (e.g., O'Dowd et al. 2004; Prather et al. 2013), I am wondering how well the used apparatus to generate the SSA, reproduces the proper size distribution of the natural SSA. Also, the authors only provided the particle size distribution and the chemical characterization for particles smaller than 500 nm and 1000 nm, respectively. There is a big problem here because while the chemical analysis was performed for submicron particles only, but the INP concentration took into account total suspended particles. It has been shown that the aerosol particles with the highest potential to act as INP are those larger than 500 nm (DeMott et al. 2010), and especially the super-micron particles (Mason et al. 2015; Gong et al. 2020) ignored in the present study. 5. INPSSA was normalized by the particle surface area. However, the NPSSA were derived from samples with total suspended particles, but the particle surface area was derived from particles ranging between 10 and 500 nm only. There is a big big mismatch here that can hide important information or can even conduct the authors to deliver wrong conclusion. That is why the following was found: "no statistically significant correlations were seen between total submicron particle counts or total SSA surface area and INPSSA at all three temperatures". Actually, it would have been more appropriate to use the size distribution of super-micron particles to calculate the particles surface area. 6. It is unclear if the SSW samples are really superficial waters (as defined by the authors), bulk waters, or deep waters. I could not find the depth at which those samples were collected. Also, the SSA was generated from waters collected at 5 m depth. Would not have been more relevant to use superficial waters instead? How comparable are the ice-nucleating abilities of the SSA particles (from "deep" waters, 5 m) with those from the superficial SML and SSW samples? 7. Section 4, the most important, is extremely short and too general without the required information to follow it. Two parametrizations were developed, one for temperatures above -22°C and one for temperatures below -22°C. Therefore, this means that the INPSSA were included in both parametrizations as the INP concentration for temperatures between -18°C and -25°C were obtained for the SSA samples. However, as mentioned above the chemical analysis for the SSA samples was performed for submicron particles only.

Therefore, this parametrizations may be valid for submicron SSA particles only and are not representative for marine aerosol particles.

---

## Author Comment (AC1) · 2 Oct 2020

We thank the reviewers for carefully reading our manuscript and for their thoughtful responses. The recommendations they gave were very valuable and have helped us to improve the paper. We have made many changes to the paper per the reviewer's requests. Notably, we added information on the comparison of our results to the literature and proposed additional parameterizations for an easier use from the community. However, the conclusions and main message of the paper did not change.

[Figure]

Our detailed point by point answer is submitted as a pdf file

Please also note the supplement to this comment:
https://acp.copernicus.org/preprints/acp-2020-487/acp-2020-487-AC1-supplement.pdf

[Figure]

**Supplement:**

We thank the reviewer for carefully reading our manuscript and for their thoughtful responses. The recommendations they gave were very valuable and have helped us to improve the paper. We have made many changes to the paper per the reviewer's request. Notably, we added information on the comparison of our results to the literature and proposed additional parameterizations for an easier use from the community. However, the conclusions and main message of the paper did not change.

Before proceeding to specific comments, we first will describe the changes made to the calculation of surface area normalized INP concentrations, as this is the basis for the rest of the changes to the manuscript.

First, we calculated adjusted daily mean particle size distribution based on sampling time intervals from the differential mobility particle sizer (DMPS) that aligned better with when the filters later analyzed by the Dynamic filter processing chamber (DFPC) were collecting particles. In our original manuscript, daily means of DMPS data were calculated on a 24-hour time interval beginning and ending at midnight. As DFPC filter samples were not collected at these exact times, there existed a small misalignment between DMPS and DFPC sampling intervals. We therefore re-calculated the DMPS daily mean across each DFPC sampling period. We also did the same adjustment for daily means of underway data when comparing underway data to the DFPC INP concentrations. We added error bars to represent the standard deviation throughout each sampling period to the resulting size distributions, produced from the bubbling system during each DFPC sampling period, shown in Figure R1. This figure has been added to the supporting information as Figure S3 on line 18.

[Figure]

**Figure R1.** Average size distributions of SSA produced by the plunging apparatus as observed by DMPS across each DFPC sampling period. Error bars represent standard deviation.

Both reviewers expressed concern that the DMPS data used to calculate surface area of SSA that did not include particles above 500 nm in diameter. The reviewers correctly pointed out that an additional mode at 800 nm exists, which contains a large portion of SSA surface area. Ovadneveite et al. (2014) developed a sea spray aerosol source function consisting of 5 log-normal modes based on in-situ particle number concentration measurements at Mace Head and open-ocean eddy correlation flux measurements from the Eastern Atlantic. Comparison of parameters from their fit with those from the fit of our number-size distribution revealed good agreement between the two. The parameters are shown in Table R1 below. This table has been added to the SI on line 1 as Table S1.

**Table R1.** Lognormal parameters for a sea spray source function parameterization from Ovadneveite et al.
(2013) and for the fit of observed particle counts during the PEACETIME cruise. For each mode (i), a geometric
standard deviation ($\sigma_i$), count-median diameter ($CMD_i$), and total number flux ($F_i$) or amplitude is shown. For
the fit from the literature (Ovadnevaite et al., 2014). $F_i$ is a function of Reynolds number $Re_{Hw}$ which we
selected as $3.1x10^6$ based on the air flow across the surface of the water in our bubbling apparatus.

| i | $\sigma_i$ | $CMD_i$ | $F_i$/Amplitude |
|---|---|---|---|
| **Ovadneveite et al. (2013)** | | | |
| 1 | 1.37 | 0.018 | $104.5(Re_{Hw} - 1x10^5)^{0.556}$ |
| 2 | 1.5 | 0.041 | $0.0442(Re_{Hw} - 1x10^5)^{1.08}$ |
| 3 | 1.42 | 0.09 | $149.6(Re_{Hw} - 1x10^5)^{0.545}$ |
| 4 | 1.53 | 0.23 | $2.96(Re_{Hw} - 1x10^5)^{0.79}$ |
| 5 | 1.85 | 0.83 | $0.51(Re_{Hw} - 1x10^5)^{0.87}$ |
| **PEACETIME Cruise** | | | |
| 1 | 1.5 | 0.01 | 0.01 |
| 2 | 1.75 | 0.035 | 0.025 |
| 3 | 1.7 | 0.115 | 0.031 |
| 4 | 1.4 | 0.300 | 0.01 |

We next took the ratio of mode 5 to mode 3 from the Ovadnevaite (2014) fit and applied it to our fit to
calculate a fifth mode accounting for particles ranging in size between 500 nm and 10 µm. Figure R2
shows an example of the result of this process using daily mean data from March 18. The total fit is
shown in gray, which consists of modes 1-4 as calculated from our DMPS data, as well as mode 5
calculated as described above. Blue circles represent observed values.

[Figure]

**Figure R2.** Example of resulting size distribution fit based on comparison of fit from observed PEACETIME
particle coutns with a 5 lognormal-mode fit from the literature (Ovadneveite et al., 2014). Blue markers denote
particle counts by the DMPS instrument (named Scanotron). Modes 1-4 are fit based on onserved data. Mode 5
is calculated by taking the ratio of Mode 5/3 from the Ovadneveite et al. (2014) fit and applying it to our
observed mode 3.

We applied this calculation to the mean data from the DMPS for each DFPC sampling period. From
the resulting fits, we calculated aerosol surface area distribution, shown in Figure R3 (also found on
line 22 of the Supporting Information as Figure S4). Finally, we used this adjusted surface area value
to re-calculate surface area normalized INP concentrations. We have added description of this
calculation to the main text on line 172.

Where relevant throughout the remainder of this text, we will refer readers to this initial comment.

[Figure]

**Figure R3.** Daily average of adjusted SSA surface area distributions. Sampling time is indicated in red text at
the top of each plot, where numbers indicate the day of the month and D/N indicates whether sampling was
conducted at day/night, respectively. The gray line shows the combined fit of modes 1-4 from observed data
with the additional contribution of mode 5 as calculated using the Ovadnevaite et al. (2013) fit . Red circles
represent observed values and blue line represents the surface area from observed values through 500nm plus
theoretical contribution from mode 5 from the gray fit. The small difference between blue and gray lines
indicates the goodness of the fit.

General Comments: The paper Trueblood et al. 2020 is a nice study which considers INP data from
oligotrophic/Mediterranean waters and shows that eutrophic parameterisations (W15 and MC18)
result in over-prediction. The occurrence of a dust deposition event over the measurement periods, in
conjunction with measurements from the SML, SSW, and SSA, makes for very interesting reading,
although it is a shame that the dataset ends before INPSSA concentrations reached a clear maximum.
However, this brings up the question can the two-component temperature dependent parameterisation
from this study be relevant to much larger bodies of water? I am happy that the authors themselves
addressed this in the need for future work relating INPSSA to POC and NCBL measurements in the
Southern Ocean. However, the difficulty in choosing POC and NCBL in relationship to INP is that all
variables must be directly measured.

We now propose a new parameterization based on OC and WIOC in SSA, which is more easily
measurable or predictable.

The authors give no indication how to apply this parameterisation in a global model.

POC classes can be retrieved from satellite data (Rasse et al., 2017) or from Biogeochemical models
such as PISCES (Aumont et al., 2015). SSA OC and WIOC characteristics can be taken from existing
parameterizations and observations (Albert et al., 2010). This information is now added to line 528 of
the manuscript.

The largest problem with the current study is that no uncertainties or error in the INP measurements
(or biological measurements for that matter) are shown or discussed. This paper should not be
published without the addition or evaluation of the inherent errors and uncertainties in the
measurements themselves and the application of the measurements to creating a parameterisation.

As mentioned in the initial comment above, we have included error bars for particle size and surface
area distributions. We have also included error bars for data from the DFPC and LINDA instruments.
See relevant sections below for details and Figure 1 on line 180, Figure 2 on line 203, and Figure 6 on
line 305 of the main text.

Also, the authors have not convincingly shown that the temperature dependent parameterisations are
necessary to model INP concentrations, although they have shown that oligotrophic waters may need
different parameterisations to eutrophic waters.

To ensure selection of the model that best fits the data, we formulated various
parameterizations consisting of different time periods, features, and number of components for
temperature ranges. Predictor features were chosen based upon their correlation with INP
concentrations as described in the previous section. Single component parameterizations in which INP
across all three temperatures  were linked with the same features were compared with two-component
parameterizations in which INP were split into warm and cold categories, each having their own
predictor features. Finally, we developed and compared altered versions of the W15 and MC18
models to account for the oligotrophic seawater of the Mediterranean Sea, as the existing models were
formulated from observations of eutrophic waters. Each parameterization was recalculated using data
across all days of the cruise as well as for only days before the dust deposition event in order to
determine the impact of the dust event on the ability to predict INP. The complete set of
parameterizations and their associated fit metrics ($R^2$ and $R_{adj.}^2$) are given in Table R2.

Figure R4a shows observed vs predicted $INP_{SSA}$ for the W15 model, while Figure R4b shows
the same but using the MC18 parameterization. Similar to our results for seawater INP, a large
overprediction is found relative to our observations when using W15. Figure R4b shows that while
MC18 is a slight improvement over the W15 approach, it still overpredicts INP by two orders of magnitude. We also present re-calculated best-fit-lines to data using the same features as in W15 and
MC18 (i.e., OC and SSA surface area) in order to account for possible changes due to the oligotrophic
nature of the Mediterranean Sea. We term these two parameterizations the altered Wilson fit for
oligotrophy, which is given by:

$$\frac{INP}{m^3} = \exp\left(-7.332 - (0.2989 * T) + (0.3792 * OC_{SSA})\right)$$

and the altered McCluskey fit for oligotrophy, given as:

$$\frac{INP}{\mu m^2} = \exp\left(-26.57 - (0.2782 * T)\right)$$

The results for these fits are shown in Figure R5a,b alongside the results of the original W15 and
MC18 parameterizations. Both altered models offer improvements over the original
parameterizations. The adjusted $R^2$ of the altered Wilson fit for oligotrophy on log-transformed INP
abundance was $R_{adj}^2=0.59$ and was $R_{adj}^2=0.32$ for the altered McCluskey fit for oligotrophy.
Interestingly, the adjusted Wilson fit for oligotrophy performs better than the adjust McCluskey fit for
oligotrophy, which is the opposite of what was found when comparing the original models.

[Figure]

**Figure R54 Different parameterizations for prediction of INP in SSA. a) W15 and refit of same method using**
**PEACETIME observations b) MK18 and refit of same method using PEACETIME observations c) single-component**
**parameterization for INP/μm² SSA surface area where INP at all temperatures are related to POC$_{SSW}$ d) two-**
**component parameterization for INP/m³ where INP≥-22°C are related to OC and INP <-22°C are related to WIOC.**

We also tried a range of novel parameterizations based on the observed correlations between
$INP_{SSA}$ with seawater and SSA properties. Below we describe two parameterizations which offered
good fits to the data. The single-component parameterization assumes the abundance of INP per unit
surface area of total SSA at each temperature can be predicted from $POC_{SSW}$ concentrations:

$$\frac{INP}{\mu m^2} = \exp(-28.5324 - (0.2729 * T) + (0.0361 * POC_{SSW}))$$

The second parameterization separates INP into warm and cold classes, where warm INP ($\geq$-
22°C) are related to SSA OC and cold INP (<-22°C) are related to the concentration of SSA WIOC.
This two-component parameterization predicts the concentration of $INP/m^3$ through the following

$$\frac{INP_{T \geq -22°C}}{m^3} = \exp\left(-7.9857 - (0.3178 * T) + (0.4643 * OC_{SSA})\right)$$

$$\frac{INP_{T < -22°C}}{m^3} = \exp\left(-6.6606 - (0.2712 * T) + (0.5755 * WIOC_{SSA})\right)$$

equations:

Figure R4c,d shows the results of our single-component model using $POC_{SSW}$ and the two-part model
which uses SSA WIOC and OC and considers the separate temperature classes of INP. The adjusted
$R^2$ for each model on the log-transformed INP abundance were $R_{adj}^2$=0.404 for the single component
model using $POC_{SSW}$ and $R_{adj}^2$=0.60 for the two-component model using OC and WIOC. This result
reveals that they both fit the observations better than the altered McCluskey parameterization for
oligotrophy, while the two-component method performs as well as the altered Wilson
parameterization. Each parameterization's fit to the data is improved when considering pre-dust days
only ($R_{adj}^2$=0.63 for the two-component parameterization and $R_{adj}^2$=0.57 for the single-component
parameterization). The improvement is more pronounced for the single-component parameterization
using $POC_{SSW}$, further pointing to the fact that such dust deposition events can alter the INP properties
of surface waters and the subsequent SSA, either through

.

**Table R2.** Summary of tested parameterizations to the PEACETIME dataset.

| Model Name | INP Units | Days | # Cat. | Features | Warm Features | Cold Features | $R^2$ | $R_{adj}^2$ |
|---|---|---|---|---|---|---|---|---|
| PD-2TC_OC_WIOC | $INP/m^3$ | Pre-Dust | 2 | | $OC_{SSA}$ | WIOC | 0.66 | 0.63 |
| PD-1TC_OC | $INP/m^3$ | Pre-Dust | 1 | $OC_{SSA}$ | | | 0.63 | 0.61 |
| PD-1TC_WSOC_WIOC | $INP/m^3$ | Pre-Dust | 1 | WSOC, WIOC | | | 0.64 | 0.60 |
| AD-2TC_OC_WIOC | $INP/m^3$ | All Days | 2 | | $OC_{SSA}$ | WIOC | 0.63 | 0.60 |
| AD-T1C_OC | $INP/m^3$ | All Days | 1 | $OC_{SSA}$ | | | 0.61 | 0.59 |
| PD-2TC_POC_PHYTO-L | $INP/\mu m^2$ | Pre-Dust | 2 | | $POC_{SSW}$ | Micro-NCBL | 0.62 | 0.59 |
| AD-1TC_WSOC_WIOC | $INP/m^3$ | All Days | 1 | WSOC, WIOC | | | 0.62 | 0.58 |
| PD-1TC_POC | $INP/\mu m^2$ | Pre-Dust | 1 | POC | | | 0.59 | 0.57 |
| PD-1TC_POC_PHYTO-L | $INP/\mu m^2$ | Pre-Dust | 1 | POC, Micro-NCBL | | | 0.58 | 0.53 |
| PD-2TC_WSOC_WIOC | $INP/m^3$ | Pre-Dust | 2 | | WSOC | WIOC | 0.53 | 0.49 |
| AD-2TC_WSOC_WIOC | $INP/m^3$ | All Days | 2 | | WSOC | WIOC | 0.45 | 0.41 |
| AD-1TC_POC | $INP/\mu m^2$ | All Days | 1 | $POC_{SSW}$ | | | 0.43 | 0.40 |
| AD-2TC_POC_PHYTO-L | $INP/\mu m^2$ | All Days | 2 | | $POC_{SSW}$ | Micro-NCBL | 0.43 | 0.39 |
| AD-2TC_POC_PHYTO-LM | $INP/\mu m^2$ | All Days | 2 | | $POC_{SSW}$ | Micro-,Nano-NCBL | 0.43 | 0.38 |
| AD-1TC_T | $INP/\mu m^2$ | All Days | 1 | Temperature | | | 0.33 | 0.32 |

We have added this discussion to line 411 of the manuscript.

A question also arises of whether INPSSA increases after the dust event are really to do with the dust
event or not? INPSSA did not seem to be very connected with SML conditions (which surprised the
authors and may therefore necessitate more attention).

A more in depth discussion of the relationship between INP concentrations and the dust event has
been added to the manuscript. See line 264 of the main text.

Lastly, throughout the text, Figure and Table descriptions are kept too short and often do not fully
describe what is shown.

We have corrected the captions related to figures and tables. Details are seen in the relevant sections
below.

Specific Comments: Temperature nomenclature (TM) varies throughout the text, sometimes for
example as -15C or -15 C. Please keep consistency and it is suggested to use the proper format of e.g.
-15∘C. All figures appear blurry, this should be corrected.

We have corrected this throughout the text.

Line 102 – SSW properties were obtained from two depths 20 cm and 5 m, why this is done at two
depths is never explained. It is important as POC is measured at 5 m depth while SML and 20 m depth
SSW samples were measured simultaneously and both calculated NCBL.

This question is linked to the question regarding Line 204 (see below). SSW properties were measured
at two depths because multiple analysis methods were available. The first method was an underway
system that continuously monitored 5 m water with a high time resolution. The second method was a
workboat which was used to collect discrete samples both the SML and the underlying seawater (at 20
cm, not 20 m). By measuring SSW properties from multiple methods (i.e., the workboat and underway),
we were able to compare results from the two and be sure of the results. Figure R5 below shows that
there was reasonable agreement between the two SSW sampling methods. Larger phytoplankton species

[Figure]

(i.e., microphytoplankton) showed greater variability between the two methods than did smaller species
(i.e., picophytoplankton), with workboat measured microphytoplankton values at times higher than
those measured from the underway. Additionally, after May 25, the underway system stopped
monitoring microphytoplankton.

**Figure R5. Daily average of continuous NCBL measurements from the underway (UWAY)**
**system, where error bars represent standard deviation compared with discrete daily samples**
**from the workboat.**

Line 110 – Why is there a specific empirical relationship for PEACETIME? Will this affect other
estimations of POC used for the parameterization?

POC was determined both continuously using optical methods and on discrete samples via high
performance liquid chromatography. The discrete samples were then used to calibrate the optical
determination of POC, as optical proxies have been found to vary from one region to another (Cetinić
et al., 2012).

Line 126 – methodology should be described in brief, or else simply cited if it is the only established
measurement practice.

We make reference to the method as described in the literature (Tovar-Sanchez 2019), on line 124 of
the main manuscript.

Line 140 – calculation should be described in brief. What are the associated errors/uncertainties of
this methodology using LINDA?

The calculation for INP from the LINDA instrument follows Stopelli (2014) which was originally
formulated by Vali (1971):

$$\frac{INP}{volume} = \frac{\ln(N_{total}) - \ln(N_{unfrozen})}{V_{tube}}$$

where $N_{total}$ is the total number of tubes, $N_{unfrozen}$ the total number of unfrozen tubes, and $V_{tube}$ the
volume of sample in each tube. The number of unfrozen tubes is calculated by first blank correcting
the number of frozen tubes, and then subtracting that value from the total number of tubes.

We calculate uncertainty as the binomial proportion confidence interval (95%) using the Wilson score
interval.

This information has been added to the main text on line 134.

Line 153 – You talk about bin size or 100-500 nm, but what is this? Is it the dry particle (electrical?)
mobility diameter? This must be stated explicitly.

For particle size distributions, we used a custom-made system referred to as Scanotron, which consists
of a DMPS and a size segregated cloud condensation nuclei counter system in parallel. The Scanotron
measures dry particle electrical mobility diameter. Data is inverted with the szdist algorithm
developed at LaMP and available online (https://hal.archives-ouvertes.fr/hal-01883795). The
inversion assumes a theoretical transfer function for the differential mobility analyzer (DMA)
and considers the condensation particle counter (CPC) efficiency and the charge equilibrium state. It
also includes multiple charge correction and accounts for diffusion losses in the instrument.
Data quality is regularly checked during inter-calibration procedures and inter-comparison
workshops, initially conducted in the frame of the EUSAAR 210 project (European Supersites
for Atmospheric Research) and since 2011 within the ACTRIS project (Wiedensohler et al.,
2012).

We have added the information regarding particle diameter to line 175 of the main text.

Line 161-164. Confusing description of how measurement of WSOC was measured vs how TOC was
measured. Then how was WIOC measured?

WSOC was measured after water extraction using a high-temperature catalytic oxidation instrument
(Shimadzu; TOC 5000 A). TOC was measured using a Multi N/C 2100 elemental analyzer (Analytik
Jena, Germany) with a furnace solids module. The analysis was performed on an 8 mm diameter filter
punch, pre-treated with 40 µL of $H_3PO_4$ (20% v/v) to remove contributions from inorganic carbon.
WIOC was determined as the difference between TOC and WSOC.

We have added this to line 155 of the main text.

Line 166-175. It seems that no measurements of ambient INP were taken. This seems concerning as
often tank and ambient measurements do not always compare well to one another. Do you have
evidence that the plunging jet SSA measurements were similar to that of the ambient SSA over the
Mediterranean?

Our goal in this experiment is to determine the contribution of INP to sea spray aerosols. As ambient
sources are expected to contain additional aerosols beyond sea spray, our bubbling setup was
necessary in order to restrict our analysis. The characteristics of the setup were selected to mimic
Fuentes et al. (2010). These parameters (water flow rates, plunging water depth, etc.) have been
shown to mimic well nascent SSA. Using this setup, our group has previously effectively mimicked
the SSA size distribution of nascent SSA (Schwier et al., 2015; 2017). Furthermore, our distribution
matches well with modes 1-4 of Ovadnevaite et al. (2014) (see initial comments at top of this file).

We have added this information to line 142 of the main text.

Line 166-175. What are the associated errors/uncertainties of this methodology using DFPC?

During an intercomparison study of the DFPC with other INP measurement systems (DeMott et al.,
2018) the DFPC was found to have uncertainties for temperature and water supersaturation of about
0.1 °C and 0.02%, respectively, leading to an overall INP concentration uncertainty of ±30%.

We have thus added 30% error bars on the observations of DFPC measured INP (Figure 1B on line
189 in the main text). We also now note this uncertainty in line 172 of the main text. For greater
explanation of the DFPC as well as how a description of its use in other studies, see our response to
reviewer 2.

Line 178 – 183. How are INP from SSW measured (I assume it is LINDA – but this is not included in
your methodology)? Which SSW measurement is tested for INP?

Correct, it is LINDA. The test is for SSW water from the workboat. We now make note of this in line
129 of the main text.

Line 191 – 192. The use of the term 'peak' here is a bit confusing in two ways. Purely graphically it is
true that INPSSA,-25C peaked on May 12, however, the implication that it is truly peaking is false as
this is the first measurement it could have been higher before measurements commenced. Has
contamination of the plunger tank system been ruled out as it is by far the greatest disparity between
different temperatures for INPSSA?

Data point on May 13 (erroneously reported as May 12) has been corrected. See initial comment on
changes to SSA surface area and averaging intervals. By correcting sampling intervals, the peak on
May 13 has been corrected. See Figure 2b in the text on line 189.

Regarding potential contamination: the plunging jet system was cleaned at the same time as the ship's
underway system and the comparison of the biological measurements from the underway seawater
system show agreement with workboat samplings, indicating no contamination across the voyage.
The plunging jet systems were additionally cleaned every day for being used in discrete seawater
generation experiments. Generated SSA concentrations were found correlated to the
nanophytoplankton cell number concentration measured online from the underway seawater system
(Sellegri et al., 2020 in review) indicating no contamination of the plunging jet system itself.

We have also altered the text describing the INP timeseries starting on line 175.

Line 196. Again the use of the word peak is a bit misleading as measurements ended before the true
peak could be observed. In this case can you really comment on the time difference between one peak
in SML and SSA?

See response to the comment above on line 191.

Line 204. Were there any differences in cell counts between SSW at 5 m or 20 cm depth?

Overall, the agreement was reasonable between SSW at 5 m and 20 cm depth. See response above to
comment on line 143 of this file and associated Figure R5 for a comparison of cell counts from
underway vs workboat.

Line 206-207. Are these the ranges associated with Pujo-Pay et al. 2011, or the ranges for this study?
If the latter than perhaps give the expected range as well.

These are ranges associated with this study. Pujo-Pay gives range of 45.3-72.4 uM for DOC and 0.80-
8.70 for POC. More specifically, Western Basin DOC ranges between 45.3-69.4 with mean of 58.7
and sigma of 7.4. Eastern Basin ranges between 49.4-72.4 with average of 61.5 and sigma of 5.9.
Western Basin POC ranges between 1.45-8.70 with mean of 4.31 and sigma of 1.73 while Eastern
Basin POC ranges between 0.80-5.41 with mean of 3.08 and sigma of 0.90.

We have added the following:

*Observed DOC and POC values ranged between 700-900 μgC/L and POC between 42-80 μgC/L and*
*were within the range of expected values for the oligotrophic Mediterranean (540—860 μgC/L for*
*DOC and 9.6-104 μgC/L for POC)(Pujo-Pay et al., 2011).*

Line 209. How are you calculating enrichment factor? It is good to state as sometimes confusion
arises.

Enrichment factor is calculated as the ratio of SML to SSW:

$$EF = \frac{SML}{SSW}$$

We have added this to line 129 of the main text.

Line 236-240. This paragraph feels like it is out of place as a discussion paragraph crammed between
the synopsis of the results in the same Figure. It does not add much to the discussion. What do these
two studies mean for your results? If anything they imply that you must compare INPSSA to SSA
bacterial abundance.

See response to the comment below.

Line 250. DOC EF is positively correlated with INPSML,-15C, and you state this is due to the dust
event, and in the next statement say that the fraction DOC enriched in the SML during the dust event
has specific IN properties. It seems possible that the DOC came from non-marine originating bacteria and that the deposition event also deposited terrestrial DOC which is the origin of increased IN
ability. Or more so, could the correlation be coincidental with another correlating factors from the
dust event (i.e. Fe)? No indication is given of why the authors believe it to be 'likely connected to the
CSP abundance, albeit not to the TEP', which if given may add value to the statement.

*We have altered the text to the following, which can be found on line 275:*

*"Figure 4 shows scatterplots of statistically significant relationships between $INP_{SML,-15C}$*
*concentrations and various SML properties. $INP_{SML,-15C}$ were most strongly positively correlated with*
*dissolved iron (r=0.99), TEP EF (r=0.95), and bacteria EF (r=0.93). However, these relationships*
*are skewed by the outlier due to the drastic increase in iron observed on June 4 (Figure S2a) from the*
*dust deposition event, as described previously. It is difficult to segregate between the dust and*
*biological impact on the $INP_{SML,-15C}$, as dust is known to have good INP properties while being*
*capable of fertilizing the surface ocean with dissolved iron, leading to concomitant increases in*
*biological activity. It is also possible that the dust deposition led to increased abundance of terrestrial*
*OC, which would exhibit different INP activity. When considering days before the dust event, $INP_{SML,-15C}$*
*$_{15C}$ is only significantly correlated with dissolved iron (r=0.91) and TOC in the SML (r=-0.93). We*
*note that while no longer statistically significant for pre-dust days, moderate correlations were still*
*observed between $INP_{SML,-15C}$ and total NCBL (r=0.48), HNA bacteria (r=0.78), and total bacteria*
*(r=0.64). Previous reports examining the correlation between INP and microbial abundance have*
*yielded mixed results. For example, a report of INP in Arctic SML and SSW found no statistically*
*significant relationship between the temperature at which 10% of droplets had frozen and bacteria or*
*phytoplankton abundances in bulk SSW and SML samples (Irish et al., 2017). However, recent*
*mesocosm studies using nutrient-enriched seawater found that INP abundances between -15°C and -*
*25°C in the aerosol phase were positively correlated with aerosolized bacterial abundance*
*(McCluskey et al., 2017). "*

Line 291-293. What are the total particle counts referred to in Line 292? How are they measured and
how do they match well with SSA counts in the range? In terms of SSA surface area: (1) how was
SSA calculated from Dp? (2) SSA have two noticeable modes larger than 500 nm, one is a submicron
mode and the other is the jet-drop mode which are found to have mean dry mobility diameters near at
0.83 (Ovadnevaite et al. 2014) and ~2 µm (Wang et al. 2017, Lewis & Schwartz 2004), respectively.
According to Figure S3, most of the surface area distributions have already peaked by 0.5 µm particle
diameter (with the possible exception of 2017-05-17), yet a significant portion of surface area for
particles with Dp> 0.5 µm seems to be lost. It seems an overstatement to say 'most of the surface area
of sea spray is comprised between this size range'. Ovadnevaite, J., Manders, A., de Leeuw, G.,
Ceburnis, D., Monahan, C., Partanen, A. I., Korhonen, H., and O'Dowd, C. D.: A sea spray aerosol
flux parameterization encapsulating wave state, Atmos. Chem. Phys., 14, 1837-1852, 10.5194/acp-14-
1837-2014, 2014. Wang, X., Deane, G. B., Moore, K. A., Ryder, O. S., Stokes, M. D., Beall, C. M.,
Collins, D. B., Santander, M. V., Burrows, S. M., Sultana, C. M., and Prather, K. A.: The role of jet
and film drops in controlling the mixing state of submicron sea spray aerosol particles, Proceedings of
the National Academy of Sciences, 114, 6978-6983, 10.1073/pnas.1702420114, 2017. Lewis, E. R.,
and Schwartz, S. E.: Sea Salt Aerosol Production: Mechanisms, Methods, Measurements and Models-
A Critical Review, American Geophysical Union, 2004.

*We refer to the initial opening comment as our response to the first portions of this comment. We do*
*not compare our particle size distribution to Wang et al. (2017) as the size distributions shown in their*
*paper are created from electrolysis bubbles, which was used to investigate the role of jet drops in*
*submicron aerosol formation. As the electrolysis bubbler created hydrogen bubbles with size less than*
*100 µm and a mean radius between 20-40 µm, no film drops would be expected to contribute to these*
*SSA since only bubbles of radius greater than 500 µm create film drops. This method would therefore*
*not be expected to accurately represent SSA. Indeed, the authors state "It is important to note that the*

nucleation bubbler is an artificial source of jet drops that was convenient to unambiguously illustrate the differences in electrical mobility between jet and film drops, but is not representative of wave breaking." Wang et al. (2017) does also show a particle size distribution for SSA generated using a plunging waterfall in a marine aerosol reference tank, but this is only for particles with diameter less than 1 µm and thus cannot be used as a reference for supermicron particle counts.

Line 313. What is the difference between SSA OC and TOC here? How is OC calculated from the

SSA?

Here, SSA OC is defined as the organic carbon content found within the aerosol phase for PM1

particles. Earlier in the manuscript this was defined as TOC, and so we will change this to make it more clear. Section 2.3.2 describes how TOC was calculated by acidifying filter punches to remove inorganic carbon, leaving only TOC.

Line 326/327. It would be good to state the relevant conclusions of Freeney et al. 2020.

We have added the following to line 406 of the main text:

*"A separate manuscript discusses the trend and controls on SSA chemical composition, linking the*

*different classes of organic carbon in submicron SSA to seawater chemical and biological properties*

*(Freney et al., 2020). In this work, OMSS was linked to $POC_{SSW}$ and the coccolithophores cell*

*abundance. In light of this and given the correlation of $INP_{SSA,-25C}$ with seawater microbial abundance*

*and with SSA OMSS and WIOC, it seems likely that $INP_{SSA}$ at this temperature are related to the*

*exudates of phytoplankton which are concentrated at the SML and then emitted into the SSA as*

*WIOC."*

Line 368. How are you calculating OMSS? Why is this in agreement with Cochran et al. 2017?

OMSS is calculated as the fraction of OM/(OM+SeaSalt), where SeaSalt is the sum of $SO_4^{2-}$, $NO_3^-$,

$NH_4^+$, $Na^+$, $Cl^-$, $K^+$, $Mg^{2+}$, $Ca^{2+}$ as determined using ICP-MS and OM is the sum of WSOM and

WIOM, which are each calculated as WSOM = WSOC x 1.8 and WIOM = WIOC * 1.4 (where

WIOC and WSOC are calculated using the method described on line 164 of this text). We have added this description to line 163 of the text.

Furthermore, we have altered the text on line 400 of the manuscript to the following:

*"Table 4 and Figure 7 shows the significant correlations between $INP_{SSA}$ and SSA properties. A positive*

*correlation exists between $INP_{SSA,-18C}$ and SSA organic carbon (OC) as well as the ratio of SSA water-soluble*

*organic carbon to organic carbon (WSOC/OC). The correlation between WSOC/OC and $INP_{SSA,-18C}$ makes*

*sense given the finding that $INP_{SSA,-18C}$ was correlated with $POC_{SSW}$, as a higher WSOC/OC value would suggest*

*a higher fraction of soluble organics which would be expected to transfer to the atmosphere from the bulk SSW*

*rather than the SML due to their high solubility. $INP_{SSA,-25C}$ had a significant correlation with WIOC and OMSS.*

*We note that $INP_{SSA,-25C}$ was also found to be correlated with various microbes in the SSW, specifically*

*Prochlorococcus, coccolithophores, nano- and micro-NCBL (previous section). Phytoplankton are known for*

*their ability to produce extracellular polymeric substances (Thornton, 2014), and a previous mesocosm*

*experiment showed microbially-derived long-chain fatty acids were efficiently ejected from the seawater as SSA,*

*increasing the fraction of highly-aliphatic, WIOC (Cochran et al., 2017). A separate manuscript discusses the*

*trend and controls on SSA chemical composition, linking the different classes of organic carbon in submicron*

*SSA to seawater chemical and biological properties (Freney et al., 2020). In this work, OMSS was linked to*

*POCSSW and the coccolithophores cell abundance. In light of this and given the correlation of INPSSA,-25C*

*with seawater microbial abundance and with SSA OMSS and WIOC, it seems likely that INPSSA at this*

*temperature are related to the exudates of phytoplankton which are concentrated at the SML and then emitted*

*into the SSA as WIOC."*

Line 413. It is stated that '. . .the INP concentrations measured in the SSW are in line with the INP

measured in the SML. . .'. There is only one comparison of INP shown of the two (figure 2a) and only one temperature is shown for the SSW. Is there further evidence to back this statement? Indicate what
evidence is referred to in the text.

This statement was vague and has been removed.

Table 1 – Description of table needs to state what p, R(R2) and n are. Is the p value of NCBL EF
0.78? This looks like a typo. Review the rest of the table to double check for other typographical
issues. Why does it say CSPabundance, when in there is no explanation of the difference between
CSP and CSPabundance?

All tables and scatter plots in the manuscript have been altered to account for these requests. We have
recalculated all correlations after calculating adjusted averaged underway values to better line up with
DFPC filter sampling time and adjusted $IN_{SSA}$ normalized by particle surface area values (explained
above). This did not impact the $INP_{SML}$ correlations with seawater properties, as daily averages were
retained. Please see Table 1 on line 262, Table 2 on line 351, Table 3 on line 360, and Table 4 on line
384.

Table 2. Description of table needs to state what p, R(R2) and n are. The table is stretched over a page
break. This should be corrected to be on one page. Change POC to POCSSW.

Please see comment above and Table 2 on line 351.

Table 3. Description of table needs to state what p, R(R2) and n are.

See comment above and Table 3 on line 360.

Figure 1. The image is blurry. The points indicated on the map are names with abbreviations that are
never explained nor referred to in the text. If these refer to the dates mentioned in other graphs, this
should be made clear. If not, then why are they there?

We have removed this figure from the manuscript.

Figure 2. Why is there no uncertainty associated with each measurement? INP measurements have
some of the largest uncertainties in aerosol science, this can't be neglected. How do you explain why
INPSSA,-25C and INPSSA,-22C are sometimes anti-correlated and sometimes not? Some other
minor corrections are needed. This graph is blurry and should be higher resolution. It would be nice to
have different keys for a) and b). The y-axis in a) should be written scientifically – i.e. either 10,000
or 1x104. It is difficult to differentiate the colours, effort should be taken to use different markers. The
bottom access should probably be the 'Date' not 'Day Number' (see same issue in other graphs).

See responses above. We have updated the figure accordingly, which can be seen on line 192 of the
manuscript.

Figure 3. This figure is also blurry with no error/uncertainty on the measurements shown.

We have updated this chart, please see Figure 4 on line 275.

Figure 4. Y-axis scale is difficult to interpret, should be written for example 108 not 108. On the x-
axis the authors might consider writing Temperature (∘C) rather than (C). Again error/uncertainties
should be shown, or else noted that the error bars are not larger than the data points. The description
of Figure 4 is on a different page than the figure, this should be corrected. It is difficult to tell
day=2017-05-24 from day=2017-06-06. The authors could probably omit the 'day=' in the key and
make the text larger.

Error bars have been added to account for INP counting errors. We included all temperature rather
than single degree averaged values. Please see Figure 6 on line 319.

Figure 5. Description does not mention INP normalised to SSA. Why use /cm3 rather than /nm2
which is what the surface area is shown in in Figure S3? When you normalise INP to SSA, should it
not still be in term of (/m3 of air  ˚u SSA cm2)? Top left panel, should read '3x10-4' not '3x10-4'.

We have updated the scatterplot figures based on this and the requests below. Please see Figures 8, 9,
and 10 on lines 355, 375, and 410 of the main text.

Figure 6. Description should be below figure, and should include some more details of the graph. The
figure is blurry, and need to be corrected. OMSS not explained.

This figure has been moved to the SI and can be found on line 27 of the SI.

Figure 7. Description should mention only significant correlations shown. Text should not state that
these panels are a matrix. The scatter plots are blurry and should be corrected to higher resolution.
The authors may choose to add r-values to each panel to make it easier for readers to study the results.

See comments above.

Figure 8. Graph should be made larger and enhanced to be less blurry. Y-axis scale is difficult to
interpret, should be written for example 101 not 101. It is difficult to read the axes. Your 3 panel axes
seem to be in different units, some per L and some per m3. These are all SSA INP so they should be
terms of their atmospheric concentration. This should be explained in the description. Additionally, it
seems clear from the graphs that while both the W15 and MC18 models over predict INP
concentrations the over prediction is not really temperature dependent. The graph seems to show more
of the difference between oligotrophic waters and eutrophic waters. How much does the authors' own
parameterizations differ if only the colder (eq. 2) or warmer (eq. 1) parameterization is applied to all
the results? Are there any data of eutrophic waters which suggest a temperature dependence might
improve the agreement?

Please see response on line 81 of this file.

Supplementary Info – consider adding a schematic of measurements taken from the tank.

Table S1. Usually tables come before Figures. Description of table needs to state what p, R(R2) and n
are. Place a '0' before all values in column p.

This table has been removed.

Figure S1. Where possible, missing data should be deleted rather than shown as a line jumping from
the last measured point to the next. There should be graph panel specific keys as each factor is not
shown on every graph. It would be nice if more detail could be given in the description of where/how
these measurements were taken. A description of what POC or biovolume covers here could also be
useful.

Figure S1 has been updated and is found on line 11 of the SI.

Figure S2. Grey outline squares around a) and b) are somewhat off centre and cut-off the a) and b). Fe
axis should be shown on the same scale in a) and b). It would be nice to see INPSSA measurement
overlaid in time with those SML and SSW conditions considered to be contributing most prominently
to INPSSA concentrations.

Figure S3. It is nearly impossible to tell some of these 'variable' apart as the same color is used for
multiple days. Please graph in such a way that the surface area spectrums can be identified for each
variable. If they are daily averages than the stdev should also be graphed. Y-axis, change from
'(nm^2/(cm^3))' to '(nm2/cm3)'. The authors could probably omit the 'variable=' in the key. Also, it is low resolution. What is a scanotron? Were these not measured by the DMPS as stated in the
methodology?

Please see Figure S3 of the supporting information on line 19.

Figure S4. Color of 'variables' again overlap for multiple days. Please graph in such a way that the
number size distribution spectrums can be identified for each variable. If they are daily averages than
the stdev should also be graphed. The y-axis shows dN/dlogDp in '(particles/(cm^3 nm))' the extra
nm is likely a typo? It should be '(/cm3)'. The authors could probably omit the 'variable=' in the key.
Also, the graph resolution is low. What is a scanotron? Were these not measured by the DMPS as
stated in the methodology?

See the answer regarding Line 153 for description of DMPS (i.e., scanotron). See Figure R1 of this
text. This has been updated in the manuscript accordingly.

Technical corrections:

We have corrected all of the concerns listed below and they are highlighted in the manuscriopt.

Line 22 – delete the 's' after INP, as INP is defined plural earlier.

Line 29 - delete the 's' after INP. This occurs many more times so check throughout the text.

Line 33 – delete extra space '. . .to SSW parameters (POCSSW. . .'. Add an 'and' or a ',' between
'(POCSSW INPSSW,-16C)'.

Line 56 – delete ')(' between references and replace with ';'. Delete '-' after SSA.

Line 62/63 – refer to study simply as 'Wilson et al. 2015 identified a temperature-dependent. . .'.
Either delete the 's' from the end of the word entities or from concentrations.

Line 68 – TM (see specific comments).

Line 85 - delete the 's' after INP. Delete the 'the' before title of study. Here is it the title of the cruise
or study? I suggest replace the word 'cruise' with 'study' and delete 'study' from the end.

Line 87 – add space, 'May 10 – June 10, 2017'. Line 88 – delete 'were'.

Line 92 – what is 'R/V'? Here 'Pourquoi Pas?' is written differently than later. Keep consistency.

Line 94 – replace 'fashion from 35◦ to 42◦ ' to 'fashion between 35◦ to 42◦ '.

Line 109 – HPLC acronym not explained.

Line 113 – FWS and SWS acronym not needed as never used again.

Line 124 – ICP-MS acronym not explained. Replace with full title as acronym not needed.

Line 130 – MQ acronym not explained. Replace with full title as acronym not needed.

Line 131 – add space between '0.5L'

Line 132 – HCL acronym not explained, although it is well known as Hydrochloric acid. Authors may
choose to spell it out as it is not repeated.

Line 135 - add space, 'May 22 - June 7'.

Line 137 – TM (see specific comments).

Line 146 – change meter to 'm'

Line 147 – ACSM acronym not described. DMPS and CPC acronym used before description.

Line 153 – correct to '10-500 nm'.

Line 159 – MSA acronym not explained. Replace with full title as acronym not needed.

Line 160 – KOH acronym not explained. Replace with full title as acronym not needed.

Line 161 – WSOC acronym used for first time and is not defined. Line 166 – '24h' change to '24-
hour' to keep consistency.

Line 167 – delete the 's' after INP, as INP is defined plural earlier.

Line 169 – change to '47 mm' with space.

Line 173 – TM (see specific comments). Add '. . .(for air temperatures of . . . -22.3 C, respectively)'.
Line 175 – add 'INP/volume of air'

Line 181/182 – TM (see specific comments). June 4 not 4th.

Line 192 – use scientific notation for INP/m3 (i.e. 1.47x10-2 not 14.7x10-3).

Line 196 – the peak in INPSSA occurred three days after INPSML peaked, not one day. Unless the
authors meant to suggest that INPSSA only saw an increase begin a full day after the INPSML peak?
Line 200 – Delete '(SI)'.

Line 204/205 – keep same scientific notation for describing cells/mL.

Line 209 – add 'Enrichment factors (EF). . .'

Line 214 – delete 'next' (optional).

Line 220 – consider adding in '. . .positive or negative correlations. . .'

Line 254 – uppercase L for litre, such that 'TOC µgC/L'. Replace 'particulate organic carbon' with
POC.

Line 255 – Replace 'dissolved organic carbon' with DOC.

Line 256 – Should be '(INP per gram of TOC)' not 'OC'. Is this cumulative INP as in W15, or is this
INP/mL?

Line 282 – Do you mean '. . .between seawater OC' or 'TOC'?

Line 291 – add space between '500' and 'nm'.

Line 294 – Only normalised size distribution shown in Figure S4, not number concentration. Perhaps
add it in the graph key? Replace 'dependence of' with 'dependence on'.

Line 298 – add space between '500' and 'nm'. Line 300 – add 'in' ahead of 'Table 2'.

Line 307 – Give correlation stats for INPSSW,-16C

Line 351 – replace 'the' with 'that'.

Line 353 – Replace 'At this C8 ACPD Interactive comment Printer-friendly version Discussion paper
temperature, INPSSA' with just 'INPSSA,-25C. . .'

Line 361 – some overlap issue with graph and line numbering.

Line 362 – change 'R=.84' to 'R=0.84'. Check for other numbering mistakes throughout the text.

Line 380/381 – TM (see specific comments).

Line 392 & equations – Warm INP defined as ≥-24C, but in eq. (1) says -22. Also, in eq. (1) 'POC'
should be rewritten 'POCSSW' to keep clarity (unless authors want any POC to be used in which case
more explanation should be given).

Line 393 – this entire line should come before eq. (1) and (2).

Line 425 –INPSWL? Change 'INPSWL and INPSML' to 'INPSSW and INPSML'.

Line 430 – Is INPSSA measured at -16C or it -18C? Leave and 'and' or ',' between POC and INP.

Line 436 – '. . .seawater POC and SSW microbial abundance' seems redundant or repetitive.

Line 446 – it is written here 'RV' but elsewhere 'R/V'. 'Pourquoi Pas ?' is also written differently
elsewhere. Please also note the supplement to this comment: https://www.atmos-chem-phys-
discuss.net/acp-2020-487/acp-2020-487-RC1- supplement.pdf

Albert, M. F. M. A., Schaap, M., de Leeuw, G., and Builtjes, P. J. H.: Progress in the determination of
the sea spray source function using satellite data, Journal of Integrative Environmental Sciences, 7,
159-166, 10.1080/19438151003621466, 2010.
Aumont, O., Ethé, C., Tagliabue, A., Bopp, L., and Gehlen, M.: PISCES-v2: an ocean biogeochemical
model for carbon and ecosystem studies, Geosci. Model Dev., 8, 2465-2513, 10.5194/gmd-8-2465-
2015, 2015.
Cetinić, I., Perry, M. J., Briggs, N. T., Kallin, E., D'Asaro, E. A., and Lee, C. M.: Particulate organic
carbon and inherent optical properties during 2008 North Atlantic Bloom Experiment, Journal of
Geophysical Research: Oceans, 117, 10.1029/2011JC007771, 2012.
DeMott, P. J., Möhler, O., Cziczo, D. J., Hiranuma, N., Petters, M. D., Petters, S. S., Belosi, F.,
Bingemer, H. G., Brooks, S. D., Budke, C., Burkert-Kohn, M., Collier, K. N., Danielczok, A., Eppers, O.,
Felgitsch, L., Garimella, S., Grothe, H., Herenz, P., Hill, T. C. J., Höhler, K., Kanji, Z. A., Kiselev, A., Koop,
T., Kristensen, T. B., Krüger, K., Kulkarni, G., Levin, E. J. T., Murray, B. J., Nicosia, A., O'Sullivan, D.,
Peckhaus, A., Polen, M. J., Price, H. C., Reicher, N., Rothenberg, D. A., Rudich, Y., Santachiara, G.,
Schiebel, T., Schrod, J., Seifried, T. M., Stratmann, F., Sullivan, R. C., Suski, K. J., Szakáll, M., Taylor, H.
P., Ullrich, R., Vergara-Temprado, J., Wagner, R., Whale, T. F., Weber, D., Welti, A., Wilson, T. W.,
Wolf, M. J., and Zenker, J.: The Fifth International Workshop on Ice Nucleation phase 2 (FIN-02):
laboratory intercomparison of ice nucleation measurements, Atmospheric Measurement
Techniques, 11, 6231-6257, 10.5194/amt-11-6231-2018, 2018.
Freney, E., Sellegri, K., Nicosia, A., Trueblood, J. V., Bloss, M., Rinaldi, M., Prevot, A., Slowik, J. G.,
Thyssen, M., Gregori, G., Engel, A., Zancker, B., Desboeufs, K., Asmi, E., Lefevre, D., and Guieu, C.:
High Time Resolution of Organic Content and Biological Origin of Mediterranean Sea Spray Aerosol,
ACP, 2020.
Fuentes, E., Coe, H., Green, D., de Leeuw, G., and McFiggans, G.: Laboratory-generated primary
marine aerosol via bubble-bursting and atomization, Atmos. Meas. Tech., 3, 141-162, 10.5194/amt-
3-141-2010, 2010.
Irish, V. E., Elizondo, P., Chen, J., Chou, C., Charette, J., Lizotte, M., Ladino, L. A., Wilson, T. W.,
Gosselin, M., Murray, B. J., Polishchuk, E., Abbatt, J. P. D., Miller, L. A., and Bertram, A. K.: Ice-
Nucleating Particles in Canadian Arctic Sea-Surface Microlayer and Bulk Seawater, Atmos. Chem.
Phys., 17, 10583-10595, https://doi.org/10.5194/acp-17-10583-2017, 2017.
McCluskey, C. S., Hill, T. C. J., Malfatti, F., Sultana, C. M., Lee, C., Santander, M. V., Beall, C. M.,
Moore, K. A., Cornwell, G. C., Collins, D. B., Prather, K. A., Jayarathne, T., Stone, E. A., Azam, F.,
Kreidenweis, S. M., and DeMott, P. J.: A Dynamic Link between Ice Nucleating Particles Released in
Nascent Sea Spray Aerosol and Oceanic Biological Activity during Two Mesocosm Experiments, J.
Atmos. Sci., 74, 151-166, https://doi.org/10.1175/JAS-D-16-0087.1, 2017.

Ovadnevaite, J., Manders, A., de Leeuw, G., Ceburnis, D., Monahan, C., Partanen, A. I., Korhonen, H.,
and O'Dowd, C. D.: A sea spray aerosol flux parameterization encapsulating wave state, Atmospheric
Chemistry and Physics, 14, 1837-1852, 10.5194/acp-14-1837-2014, 2014.
Pujo-Pay, M., Conan, P., Oriol, L., Cornet-Barthaux, V. C., Falco, C., Ghiglione, J.-F., Goyet, C., Moutin,
T., and Prieur, L.: Integrated Survey of Elemental Stoichiometry (C, N, P) from the Western to Eastern
Mediterranean Sea, Biogeosciences, 8, 883-899, https://doi.org/10.5194/bg-8-883-2011, 2011.
Rasse, R., Dall'Olmo, G., Graff, J., Westberry, T. K., van Dongen-Vogels, V., and Behrenfeld, M. J.:
Evaluating Optical Proxies of Particulate Organic Carbon across the Surface Atlantic Ocean, Frontiers
in Marine Science, 4, 367, 2017.

---

## Author Comment (AC2) · 2 Oct 2020

We thank the reviewer for carefully reading our manuscript and for providing us with their helpful insight. We have adjusted our manuscript based on their recommendations, which we believe has strengthened the manuscript's overall impact. Our point by point responses can be found as a suppl. pdf file.

Please also note the supplement to this comment:
https://acp.copernicus.org/preprints/acp-2020-487/acp-2020-487-AC2-supplement.pdf

[Figure]

**Supplement:**

We thank the reviewer for carefully reading our manuscript and for providing us with their helpful insight. We have adjusted our manuscript based on their recommendations, which we believe has strengthened the manuscript's overall impact. Our responses to specific comments are shown below in red.

Review of "A Two-Component Parameterization of Marine Ice Nucleating Particles Based on Seawater Biology and Sea Spray Aerosol Measurements in the Mediterranean Sea" by Trueblood et al.

General comment

This study investigated the ice-nucleating abilities of surface microlayer (SML), surface seawaters (SSW), and sea spray aerosol (SSA) particles collected/generated in the Mediterranean Sea. In parallel to the evaluation of the ice-nucleating abilities of the different samples, a large set of biogeochemical analysis were performed on the samples to understand the relationship between ocean biology and marine ice nucleating particles (INP). While the ice nucleation analysis of the SML and SSW samples was performed with a LED based Ice Nuclei Detection Apparatus (LINDA), the analysis for the SSA samples was performed using a Dynamic Filter Processing Chamber (DFPC). Taking into account the collected information, the authors developed a new two-component parametrization. Although the information collected/derived by the authors is very rich and valuable, in addition that they were collected in a poorly explored region on Earth, the manuscript is not easy to follows, it lacks important information, and the conclusions are not clearly supported by the provided data. The manuscript fits with the ACP scope, but the current version cannot be accepted for its publication.

Major Comment 1. Two different techniques were used to analyze the ice nucleating abilities of the samples, i.e., the LINDA and the DFPC. While the LINDA determines the INP concentrations via the immersion freezing mode, the DFPC does it via condensation freezing. Given that both data sets were used to develop the parametrization, I am wondering if the INP concentration delivered by both instruments are directly comparable. For example, I am wondering about the very low concentration of INPs reported for the SSA samples in comparison to literature data. It is a true number or is it an artifact related to the used method?

Only the DFPC IN data was used to develop the parameterization. The LINDA instrument was only used on seawater (SSW and SML) samples, and the results were used to help guide the selection of features for the parameterization of SSA INP based on their correlation with biological features. A more in-depth response regarding the process of using the DFPC and its comparison with immersion freezing methods is given in the response below.

2. I was unable to fully understand how the INP concentrations for the SSA samples were obtained as the DFPC was not properly described. Is this a new custom-made instrument? Is this the first data delivered/published by this instrument? If this is the case, a much deeper description needs to be provided. If this is not the case, how good is the agreement of the data delivered by the DFPC against other well-known ice nucleation instruments?

The method for the DFPC is as follows: aerosols are collected onto a filter which is then placed on a metallic support covered with a layer of paraffin. The paraffin is heated slightly (60-80˚C) for ~10 seconds and then rapidly cooled to allow the paraffin to penetrate the filter pores. This process is done to modify the color of the filter from white to black for optical processing and to improve thermal conductivity. The filter is then placed inside the DFPC on top of a Peltier cooler plate. The filter is then continuously swept by humid and cool air. Upon activation, ice nuclei grow into ice crystals with size around 0.4 mm. The ice crystals are then recorded using a camera.

In this study, the DFPC was used at water saturation, $S_w$, of 1.02 and T=-22, -25, and -18˚C. Under these conditions, we expect to see condensation freezing mode. It has been reported (Vali et al., 2015) that condensation freezing and immersion freezing are not distinguishable as unique.

The DFPC has been used in multiple previous studies and found to agree well with other INP monitoring instruments (DeMott et al., 2018; Hiranuma et al., 2019; McCluskey et al., 2018).

We have updated the methods section starting at line 167 of the main text to account for this added information.

3. I am surprised that the INPs concentrations of the SML, SSW, and SSA are not compared to literature data. Actually, a recent study by Gong et al. (2020) who also studied the SML, SSW, and SSA is not cited/discussed here. There are also other studies in marine environment performed at subtropical and tropical latitudes that deserved to be discussed in the context of the present study.

We have added Figure 2 and Figure 3 to the manuscript, along with discussion starting on line 222 to compare seawater and SSA INP concentrations to previous studies.

4. Given that the chemical composition of the SSA is linked to the size of the aerosol particles (e.g., O0Dowd et al. 2004; Prather et al. 2013), I am wondering how well the used apparatus to generate the SSA, reproduces the proper size distribution of the natural SSA. Also, the authors only provided the particle size distribution and the chemical characterization for particles smaller than 500 nm and 1000 nm, respectively. There is a big problem here because while the chemical analysis was performed for submicron particles only, but the INP concentration took into account total suspended particles. It has been shown that the aerosol particles with the highest potential to act as INP are those larger than 500 nm (DeMott et al. 2010), and especially the super-micron particles (Mason et al. 2015; Gong et al. 2020) ignored in the present study.

We agree that replicating the production mechanism of SSA is certainly an important and difficult task. The plunging apparatus used in this study has been well characterized in previous studies (Schwier et al., 2015; Schwier et al., 2017). Furthermore, the characteristics of the setup (water flow rates, plunging water depth, etc.) were selected to mimic Fuentes et al. (2010), which have been shown to mimic quite well the production of nascent SSA. To this end, we are confident that the SSA analyzed in this study are an accurate representation of natural SSA.

It is not yet clear whether INP from sea spray are mainly from super or submicron sizes. While Gong et al. (2020) reported high concentrations of INP for supermicron particles, they also note these likely came from dust particles rather than marine aerosols. As studies have shown that the IN active components of SSA are typically of biogenic origin (DeMott et al., 2016), and that the organic mass fraction of SSA is greatest for submicron particles (Gantt and Meskhidze, 2013), it would seem likely that the INP characteristics of SSA can be well described in terms of the submicron regime. Indeed, comparison of INP concentrations in PM1 and PM10 filter samples of both clean marine air and continental land-mass air at Mace Head in the North Atlantic have recently provided evidence that the majority of SSA INP are in the submicron regime and that terrestrial aerosol INP are in the supermicron regime (McCluskey et al., 2018). Field campaigns in the equatorial Pacific and Gulf of Mexico have also found the marine INPs are most abundant for particle diameters less than 500 nm (Rosinski et al., 1986, 1987; Rosinski et al., 1988). However, we want to stress here that our INP measurements are performed over the full range of SSA sizes, and we do not ignore its supermicronic fraction.

For a response to the concern of particle size distributions, please see our response to comment number 5 below.

Concerning the chemical analysis of SSA, it is correct that only the submicron concentration of SSA has been measured. However, previous analysis of the chemical composition of marine aerosol have shown that most of the organic matter is found in the submicron fraction (O'Dowd et al. 2004, Gantt and Meskhidze 2013 for a review). Therefore, we base our analysis that the WIOC, WSOC and OC concentrations provided in this work are representative of the whole size range of SSA. This is now mentioned line 698-700 of the manuscript.

Given that we have taken steps to accurately mimic the SSA production mechanism, extended our SSA surface calculation to include particle with diameters smaller than 10 μm, and clearly mention the hypothesis that organic carbon in submicron SSA is representative of the OC in the totality of the SSA, we are confident that our approach is sufficient to characterize total SSA INP concentrations. However, future studies are needed to better constrain the characteristics of SSA in sub and supermicron regimes.

5. INPSSA was normalized by the particle surface area. However, the NPSSA were derived from samples with total suspended particles, but the particle surface area was derived from particles ranging between 10 and 500 nm only. There is a big mismatch here that can hide important information or can even conduct the authors to deliver wrong conclusion. That is why the following was found: "no statistically significant correlations were seen between total submicron particle counts or total SSA surface area and INPSSA at all three temperatures". Actually, it would have been more appropriate to use the size distribution of super-micron particles to calculate the particles surface area.

Please see below our response to reviewer 1 who had the same concern:

First, we calculated adjusted daily mean particle size distribution based on sampling time intervals from the differential mobility particle sizer (DMPS) that aligned better with when the filters later analyzed by the Dynamic filter processing chamber (DFPC) were collecting particles. In our original manuscript, daily means of DMPS data were calculated on a 24-hour time interval beginning and ending at midnight. As DFPC filter samples were not collected at these exact times, there existed a small misalignment between DMPS and DFPC sampling intervals. We therefore re-calculated the DMPS daily mean across each DFPC sampling period. We also did the same adjustment for daily means of underway data when comparing underway data to the DFPC INP concentrations. We added error bars to represent the standard deviation throughout each sampling period to the resulting size distributions, produced from the bubbling system during each DFPC sampling period, shown in Figure R1. This figure has been added to the supporting information as Figure S3 on line 18.

[Figure]

**Figure R1.** Average size distributions of SSA produced by the plunging apparatus as observed by DMPS across each DFPC sampling period. Error bars represent standard deviation.

Both reviewers expressed concern that the DMPS data used to calculate surface area of SSA did not include particles above 500 nm in diameter. The reviewers correctly pointed out that an additional mode at 800 nm exists, which contains a large portion of SSA surface area. Ovadneveite et al. (2014) developed a sea spray aerosol source function consisting of 5 log-normal modes based on in-situ particle number concentration measurements at Mace Head and open-ocean eddy correlation flux measurements from the Eastern Atlantic. Comparison of parameters from their fit with those from the fit of our number-size distribution revealed good agreement between the two. The parameters are shown in Table R1 below. This table has been added to the SI on line 1 as Table S1.

**Table R1.** Lognormal parameters for a sea spray source function parameterization from Ovadneveite et al. (2013) and for the fit of observed particle counts during the PEACETIME cruise. For each mode (i), a geometric standard deviation ($\sigma_i$), count-median diameter ($CMD_i$), and total number flux ($F_i$) or amplitude is shown. For the fit from the literature (Ovadnevaite et al., 2014). $F_i$ is a function of Reynolds number $Re_{Hw}$ which we selected as $3.1 \times 10^6$ based on the air flow across the surface of the water in our bubbling apparatus.

| i | $\sigma_i$ | $CMD_i$ | $F_i$/Amplitude |
|---|---|---|---|
| **Ovadneveite et al. (2013)** | | | |
| 1 | 1.37 | 0.018 | $104.5(Re_{Hw} - 1x10^5)^{0.556}$ |
| 2 | 1.5 | 0.041 | $0.0442(Re_{Hw} - 1x10^5)^{1.08}$ |
| 3 | 1.42 | 0.09 | $149.6(Re_{Hw} - 1x10^5)^{0.545}$ |
| 4 | 1.53 | 0.23 | $2.96(Re_{Hw} - 1x10^5)^{0.79}$ |
| 5 | 1.85 | 0.83 | $0.51(Re_{Hw} - 1x10^5)^{0.87}$ |
| **PEACETIME Cruise** | | | |
| 1 | 1.5 | 0.01 | 0.01 |
| 2 | 1.75 | 0.035 | 0.025 |
| 3 | 1.7 | 0.115 | 0.031 |
| 4 | 1.4 | 0.300 | 0.01 |

We next took the ratio of mode 5 to mode 3 from the Ovadnevaite (2014) fit and applied it to our fit to calculate a fifth mode accounting for particles ranging in size between 500 nm and 10 μm. Figure R2 shows an example of the result of this process using daily mean data from March 18. The total fit is shown in gray, which consists of modes 1-4 as calculated from our DMPS data, as well as mode 5 calculated as described above. Blue circles represent observed values.

[Figure]

**Figure R2.** Example of resulting size distribution fit based on comparison of fit from observed PEACETIME particle coutns with a 5 lognormal-mode fit from the literature (Ovadneveite et al., 2014). Blue markers denote particle counts by the DMPS instrument (named Scanotron). Modes 1-4 are fit based on onserved data. Mode 5 is calculated by taking the ratio of Mode 5/3 from the Ovadneveite et al. (2014) fit and applying it to our observed mode 3.

We applied this calculation to the mean data from the DMPS for each DFPC sampling period. From the resulting fits, we calculated aerosol surface area distribution, shown in Figure R3 (also found on line 22 of the Supporting Information as Figure S4). Finally, we used this adjusted surface area value to re-calculate surface area normalized INP concentrations. We have added description of this calculation to the main text on line 172.

[Figure]

**Figure R3.** Daily average of adjusted SSA surface area distributions. Sampling time is indicated in red text at the top of each plot, where numbers indicate the day of the month and D/N indicates whether sampling was conducted at day/night, respectively. The gray line shows the combined fit of modes 1-4 from observed data with the additional contribution of mode 5 as calculated using the Ovadnevaite et al. (2013) fit . Red circles represent observed values and blue line represents the surface area from observed values through 500nm plus theoretical contribution from mode 5 from the gray fit. The small difference between blue and gray lines indicates the goodness of the fit.

6. It is unclear if the SSW samples are really superficial waters (as defined by the authors), bulk waters, or deep waters. I could not find the depth at which those samples were collected. Also, the SSA was generated from waters collected at 5 m depth. Would not have been more relevant to use superficial waters instead? How comparable are the ice-nucleating abilities of the SSA particles (from "deep" waters, 5 m) with those from the superficial SML and SSW samples?

We do mention the sampling depth of discrete SSW samples and underway continuous seawater (respectively 20 cm and 5 m). As the 5 m waters showed similar properties to SSW collected from 20

[Figure]

cm (see Figure R4), we are confident that they can be considered similar in nature.

**Figure R4.** Daily average of continuous NCBL measurements from the underway (UWAY) system, where error bars represent standard deviation compared with discrete daily samples from the workboat at 20 cm.

7. Section 4, the most important, is extremely short and too general without the required information to follow it. Two parametrizations were developed, one for temperatures above -22°C and one for temperatures below -22°C. Therefore, this means that the INPSSA were included in both parametrizations as the INP concentration for temperatures between -18°C and -25°C were obtained for the SSA samples. However, as mentioned above the chemical analysis for the SSA samples was performed for submicron particles only. C3 Therefore, this parametrizations may be valid for submicron SSA particles only and are not representative for marine aerosol particles.

Concerning the chemical analysis of SSA, see our answer at point 4.

Reviewer 1 rose similar concerns regarding Section 4. Please see our response below:

To ensure selection of the model that best fits the data, we formulated various parameterizations consisting of different time periods, features, and number of components for temperature ranges. Predictor features were chosen based upon their correlation with INP concentrations as described in the previous section. Single component parameterizations in which INP across all three temperatures were linked with the same features were compared with two-component parameterizations in which INP were split into warm and cold categories, each having their own predictor features. Finally, we developed and compared altered versions of the W15 and MC18 models to account for the oligotrophic seawater of the Mediterranean Sea, as the existing models were formulated from observations of eutrophic waters. Each parameterization was recalculated using data across all days of the cruise as well as for only days before the dust deposition event in order to

determine the impact of the dust event on the ability to predict INP. The complete set of parameterizations and their associated fit metrics ($R^2$ and $R_{adj.}^2$) are given in Table R2.

Figure R5a shows observed vs predicted $INP_{SSA}$ for the W15 model, while Figure R5b shows the same but using the MC18 parameterization. Similar to our results for seawater INP, a large overprediction is found relative to our observations when using W15. Figure 5b shows that while MC18 is a slight improvement over the W15 approach, it still overpredicts INP by two orders of magnitude. We also present re-calculated best-fit-lines to data using the same features as in W15 and MC18 (i.e., OC and SSA surface area) in order to account for possible changes due to the oligotrophic nature of the Mediterranean Sea. We term these two parameterizations the altered Wilson fit for oligotrophy, which is given by:

$$\frac{INP}{m^3} = \exp\left(-7.332 - (0.2989 * T) + (0.3792 * OC_{SSA})\right)$$

and the altered McCluskey fit for oligotrophy, given as:

$$\frac{INP}{\mu m^2} = \exp\left(-26.57 - (0.2782 * T)\right)$$

The results for these fits are shown in Figure R5a,b alongside the results of the original W15 and MC18 parameterizations. Both altered models offer improvements over the original parameterizations. The adjusted $R^2$ of the altered Wilson fit for oligotrophy on log-transformed INP abundance was $R_{adj}^2=0.59$ and was $R_{adj}^2=0.32$ for the altered McCluskey fit for oligotrophy.

[Figure]

Interestingly, the adjusted Wilson fit for oligotrophy performs better than the adjust McCluskey fit for oligotrophy, which is the opposite of what was found when comparing the original models.

**Figure R5. Different parameterizations for prediction of INP in SSA. a) W15 and refit of same method using PEACETIME observations b) MK18 and refit of same method using PEACETIME observations c) single-component parameterization for INP/μm² SSA surface area where INP at all temperatures are related to POC$_{SSW}$ d) two-component parameterization for INP/m³ where INP≥-22°C are related to OC and INP <-22°C are related to WIOC.**

We also tried a range of novel parameterizations based on the observed correlations between INP$_{SSA}$ with seawater and SSA properties. Below we describe two parameterizations which offered good fits to the data. The single-component parameterization assumes the abundance of INP per unit surface area of total SSA at each temperature can be predicted from POC$_{SSW}$ concentrations:

$$\frac{INP}{\mu m^2} = \exp(-28.5324 - (0.2729 * T) + (0.0361 * POC_{SSW})$$

The second parameterization separates INP into warm and cold classes, where warm INP (≥-22°C) are related to SSA OC and cold INP (<-22°C) are related to the concentration of SSA WIOC. This two-component parameterization predicts the concentration of INP/m³ through the following

$$\frac{INP_{T\geq-22°C}}{m^3} = \exp\left(-7.9857 - (0.3178 * T) + (0.4643 * OC_{SSA})\right)$$

$$\frac{INP_{T<-22°C}}{m^3} = \exp\left(-6.6606 - (0.2712 * T) + (0.5755 * WIOC_{SSA})\right)$$

equations:

Figure R5c,d shows the results of our single-component model using POC$_{SSW}$ and the two-part model which uses SSA WIOC and OC and considers the separate temperature classes of INP. The adjusted R² for each model on the log-transformed INP abundance were R$_{adj}$²=0.404 for the single component model using POC$_{SSW}$ and R$_{adj}$²=0.60 for the two-component model using OC and WIOC. This result reveals that they both fit the observations better than the altered McCluskey parameterization for oligotrophy, while the two-component method performs as well as the altered Wilson parameterization. Each parameterization's fit to the data is improved when considering pre-dust days only (R$_{adj}$²=0.63 for the two-component parameterization and R$_{adj}$²=0.57 for the single-component parameterization). The improvement is more pronounced for the single-component parameterization using POC$_{SSW}$, further pointing to the fact that such dust deposition events can alter the INP properties of surface waters and the subsequent SSA, either through

.

**Table R2.** Summary of tested parameterizations to the PEACETIME dataset.

| Model Name | INP Units | Days | # Cat. | Features | Warm Features | Cold Features | $R^2$ | $R_{adj}^2$ |
|---|---|---|---|---|---|---|---|---|
| PD-2TC_OC_WIOC | INP/m$^3$ | Pre-Dust | 2 | | OC$_{SSA}$ | WIOC | 0.66 | 0.63 |
| PD-1TC_OC | INP/m$^3$ | Pre-Dust | 1 | OC$_{SSA}$ | | | 0.63 | 0.61 |
| PD-1TC_WSOC_WIOC | INP/m$^3$ | Pre-Dust | 1 | WSOC, WIOC | | | 0.64 | 0.60 |
| AD-2TC_OC_WIOC | INP/m$^3$ | All Days | 2 | | OC$_{SSA}$ | WIOC | 0.63 | 0.60 |
| AD-T1C_OC | INP/m$^3$ | All Days | 1 | OC$_{SSA}$ | | | 0.61 | 0.59 |
| PD-2TC_POC_PHYTO-L | INP/µm$^2$ | Pre-Dust | 2 | | POC$_{SSW}$ | Micro-NCBL | 0.62 | 0.59 |
| AD-1TC_WSOC_WIOC | INP/m$^3$ | All Days | 1 | WSOC, WIOC | | | 0.62 | 0.58 |
| PD-1TC_POC | INP/µm$^2$ | Pre-Dust | 1 | POC | | | 0.59 | 0.57 |
| PD-1TC_POC_PHYTO-L | INP/µm$^2$ | Pre-Dust | 1 | POC, Micro-NCBL | | | 0.58 | 0.53 |
| PD-2TC_WSOC_WIOC | INP/m$^3$ | Pre-Dust | 2 | | WSOC | WIOC | 0.53 | 0.49 |
| AD-2TC_WSOC_WIOC | INP/m$^3$ | All Days | 2 | | WSOC | WIOC | 0.45 | 0.41 |
| AD-1TC_POC | INP/µm$^2$ | All Days | 1 | POC$_{SSW}$ | | | 0.43 | 0.40 |
| AD-2TC_POC_PHYTO-L | INP/µm$^2$ | All Days | 2 | | POC$_{SSW}$ | Micro-NCBL | 0.43 | 0.39 |
| AD-2TC_POC_PHYTO-LM | INP/µm$^2$ | All Days | 2 | | POC$_{SSW}$ | Micro-,Nano-NCBL | 0.43 | 0.38 |
| AD-1TC_T | INP/µm$^2$ | All Days | 1 | Temperature | | | 0.33 | 0.32 |

We have added this discussion to line 411 of the manuscript.

**References**

DeMott, P. J., Hill, T. C. J., McCluskey, C. S., Prather, K. A., Collins, D. B., Sullivan, R. C., Ruppel, M. J., Mason, R. H., Irish, V. E., Lee, T., Hwang, C. Y., Rhee, T. S., Snider, J. R., McMeeking, G. R., Dhaniyala, S., Lewis, E. R., Wentzell, J. J. B., Abbatt, J., Lee, C., Sultana, C. M., Ault, A. P., Axson, J. L., Martinez, M. D., Venero, I., Santos-Figueroa, G., Stokes, D. M., Deane, G. B., Mayol-Bracero, O. L., Grassian, V. H., Bertram, T. H., Bertram, A. K., Moffett, B. F., and Franc, G. D.: Sea Spray Aerosol as a Unique Source of ice Nucleating Particles, PNAS, 113, 5797-5803, https://doi.org/10.1073/pnas.1514034112, 2016.

DeMott, P. J., Möhler, O., Cziczo, D. J., Hiranuma, N., Petters, M. D., Petters, S. S., Belosi, F., Bingemer, H. G., Brooks, S. D., Budke, C., Burkert-Kohn, M., Collier, K. N., Danielczok, A., Eppers, O., Felgitsch, L., Garimella, S., Grothe, H., Herenz, P., Hill, T. C. J., Höhler, K., Kanji, Z. A., Kiselev, A., Koop, T., Kristensen, T. B., Krüger, K., Kulkarni, G., Levin, E. J. T., Murray, B. J., Nicosia, A., O'Sullivan, D., Peckhaus, A., Polen, M. J., Price, H. C., Reicher, N., Rothenberg, D. A., Rudich, Y., Santachiara, G., Schiebel, T., Schrod, J., Seifried, T. M., Stratmann, F., Sullivan, R. C., Suski, K. J., Szakáll, M., Taylor, H. P., Ullrich, R., Vergara-Temprado, J., Wagner, R., Whale, T. F., Weber, D., Welti, A., Wilson, T. W., Wolf, M. J., and Zenker, J.: The Fifth International Workshop on Ice Nucleation phase 2 (FIN-02): laboratory intercomparison of ice nucleation measurements, Atmospheric Measurement Techniques, 11, 6231-6257, 10.5194/amt-11-6231-2018, 2018.

Fuentes, E., Coe, H., Green, D., de Leeuw, G., and McFiggans, G.: Laboratory-generated primary marine aerosol via bubble-bursting and atomization, Atmos. Meas. Tech., 3, 141-162, 10.5194/amt-3-141-2010, 2010.

Gantt, B., and Meskhidze, N.: The physical and chemical characteristics of marine primary organic aerosol: a review, Atmospheric Chemistry and Physics, 13, 3979-3996, 10.5194/acp-13-3979-2013, 2013.

Gong, X., Wex, H., van Pinxteren, M., Triesch, N., Fomba, K. W., Lubitz, J., Stolle, C., Robinson, T.-B., Müller, T., Herrmann, H., and Stratmann, F.: Characterization of aerosol particles at Cabo Verde close to sea level and at the cloud level – Part 2: Ice-nucleating particles in air, cloud and seawater, Atmospheric Chemistry and Physics, 20, 1451-1468, 10.5194/acp-20-1451-2020, 2020.

Hiranuma, N., Adachi, K., Bell, D. M., Belosi, F., Beydoun, H., Bhaduri, B., Bingemer, H., Budke, C., Clemen, H.-C., Conen, F., Cory, K. M., Curtius, J., DeMott, P. J., Eppers, O., Grawe, S., Hartmann, S., Hoffmann, N., Höhler, K., Jantsch, E., Kiselev, A., Koop, T., Kulkarni, G., Mayer, A., Murakami, M., Murray, B. J., Nicosia, A., Petters, M. D., Piazza, M., Polen, M., Reicher, N., Rudich, Y., Saito, A., Santachiara, G., Schiebel, T., Schill, G. P., Schneider, J., Segev, L., Stopelli, E., Sullivan, R. C., Suski, K., Szakáll, M., Tajiri, T., Taylor, H., Tobo, Y., Ullrich, R., Weber, D., Wex, H., Whale, T. F., Whiteside, C. L., Yamashita, K., Zelenyuk, A., and Möhler, O.: A comprehensive characterization of ice nucleation by three different types of cellulose particles immersed in water, Atmospheric Chemistry and Physics, 19, 4823-4849, 10.5194/acp-19-4823-2019, 2019.

McCluskey, C. S., Ovadnevaite, J., Rinaldi, M., Atkinson, J., Belosi, F., Ceburnis, D., Marullo, S., Hill, T. C. J., Lohmann, U., Kanji, Z. A., O'Dowd, C., Kreidenweis, S. M., and DeMott, P. J.: Marine and Terrestrial Organic Ice-Nucleating Particles in Pristine Marine to Continentally Influenced Northeast Atlantic Air Masses, J. Geophys Res. Atmos., 123, 6196-6212, https://doi.org/10.1029/2017JD028033, 2018.

O'Dowd CD, Facchini MC, Cavalli F, Ceburnis D, Mircea M, Decesari S, Fuzzi S, Yoon YJ, Putaud JP. Biogenically driven organic contribution to marine aerosol. Nature. 2004 Oct 7;431(7009):676-80. doi: 10.1038/nature02959. PMID: 15470425.

Ovadnevaite, J., Manders, A., de Leeuw, G., Ceburnis, D., Monahan, C., Partanen, A. I., Korhonen, H., and O'Dowd, C. D.: A sea spray aerosol flux parameterization encapsulating wave state, Atmospheric Chemistry and Physics, 14, 1837-1852, 10.5194/acp-14-1837-2014, 2014.

Rosinski, J., Haagenson, P. L., Nagamoto, C. T., and Parungo, F.: Ice-forming nuclei of maritime origin, Journal of Aerosol Science, 17, 23-46, https://doi.org/10.1016/0021-8502(86)90004-2, 1986.

Rosinski, J., Haagenson, P. L., Nagamoto, C. T., and Parungo, F.: Nature of ice-forming nuclei in marine air masses, Journal of Aerosol Science, 18, 291-309, https://doi.org/10.1016/0021-8502(87)90024-3, 1987.

Rosinski, J., Haagenson, P. L., Nagamoto, C. T., Quintana, B., Parungo, F., and Hoyt, S. D.: Ice-forming nuclei in air masses over the Gulf of Mexico, Journal of Aerosol Science, 19, 539-551, https://doi.org/10.1016/0021-8502(88)90206-6, 1988.

Schwier, A. N., Rose, C., Asmi, E., Ebling, A. M., Landing, W. M., Marro, S., Pedrotti, M.-L., Sallon, A., Iuculano, F., Agusti, S., Tsiola, A., Pitta, P., Louis, J., Guieu, C., Gazeau, F., and Sellegri, K.: Primary Marine Aerosol Emissions from the Mediterranean Sea During Pre-Bloom and Oligotrophic Conditions: Correlations to Seawater Chlorophyll-a From a Mesocosm Study, Atmos. Chem. Phys., 15, 7961-7976, https://doi.org/10.5194/acp-15-7961-2015, 2015.

Schwier, A. N., Sellegri, K., Mas, S., Charrière, B., Pey, J., Rose, C., Temime-Roussel, B., Jaffrezo, J.-L., Parin, D., Picard, D., Ribeiro, M., Roberts, G., Sempéré, R., Marchand, N., and D'Anna, B.: Primary marine aerosol physical flux and chemical composition during a nutrient enrichment experiment in mesocosms in the Mediterranean Sea, Atmospheric Chemistry and Physics, 17, 14645-14660, 10.5194/acp-17-14645-2017, 2017.

Vali, G., DeMott, P. J., Möhler, O., and Whale, T. F.: Technical Note: A proposal for ice nucleation terminology, Atmospheric Chemistry and Physics, 15, 10263-10270, 10.5194/acp-15-10263-2015, 2015.

---

## Referee Report (RR1)

Reviewer 1 Response to Revised Submission ' A Two-Component Parameterization of Marine Ice Nucleating Particles Based on Seawater Biology and Sea Spray Aerosol Measurements in the Mediterranean Sea' by Jonathan V. Trueblood et al.

MS No.: acp-2020-487

MS type: Research article

Iteration: Revised Submission

General Comments: The paper Trueblood et al. 2020 has taken this reviewer's corrections on board and satisfactorily answered this reviewer's comments. I recommend this article for publication, pending a few minor revisions.

Specific comments for minor corrections:

Line 34 – POC acronym used without definition in abstract

Line 52 – the word 'recent' may be omitted as studies are from 1973 & 1976

Line 65 – TOC acronym used without definition for first time

Line 82-86 – Listed numbers should have commas

Line 92 – a space should be added around hyphen to read ' May 10 – June 10'

Line 167 – authors have equations with multipliers of 'x' and '*' in same sentence. Choose one for consistency.

Line 192 – it should read '0.5'

Line 197 – replace 500 nm with 0.5 μm

Line 213 – authors may use INP rather than 'ice nucleating particles'

Line 246 – it should read '…INP in ambient aerosol in future studies.'

Line 266 – the long dash should be replaced with a short dash, i.e. '-'

Line 279 (Table 1 description) – authors may replace p>.05 with 'p>0.05'. This is repeated in all table descriptions.

Line 279 – 280 (Table 1) – DOC_SSW r-val pre-dust should be italicized. Dissolved Iron p values should have a 0 in front of decimal place. Bacteria HNA r-val pre-dust should be italicized. TOC_SML r-val all days should be italicized.

Line 286 – authors report pre dust days with r = 0.83, when table states this is all days

Line 293 (Figure 4) – Reported R2 values on graphs are overlaid on data point, the authors may adjust this for better clarity.

Line 316 – authors may use TOC rather than 'total organic carbon'

Line 338 – 342 – The authors summarise results just presented which is redundant, these sentences should be deleted such that the text reads, '…presumably due to low biological productivity. This complicated relationship between seawater TOC and INP_SML…'

Line 352 – replace 'of' with 'on'

Line 355 - replace 500 nm with 0.5 µm

Line 367 – do the authors mean micro-NCBL?

Line 421 – acronym already defined, replace 'organic mass fraction od sea-spray (OMSS)' with 'OMSS'

Line 432 (Figure 9 description) – text should read 'Figure 9. Scatter plots of INPSSA at three temperatures and SSA properties, with corresponding correlation statistics s reported in Table 4...' (note- if nothing else the text incorrectly states Table 2)

Figure 469 (Figure 10 description) – 'b)', 'c)', and 'd)' should be proceeded by a period or comma

Line 490 – average should be $2.1 \times 10^5$

Line 515 – 517 – Have the authors considered a delayed biological surge or behaviour from Fe deposition?

Line 520 - acronym already defined; replace 'water soluble organic matter' with 'WSOM'

Line 522 - acronym already defined; replace 'water insoluble organic carbon' with 'WIOC'

Line 524- acronym already defined; replace 'water insoluble organic matter' with 'WIOM'. In relation to line 522, do authors mean to differentiate with WIOC and WIOM here?

Line 533 – add text to read, 'The re-calculated parametrization'

Line 534-536 – Is it the author's intent that the two-component model be used for Eutrophic waters too? If not the authors may adjust the text to read '...incorporate marine INP emissions from oligotrophic waters into numerical models.'

Supplementary text:

Table S2 – Model names in table are not explained

Figures S3 – each panel graph has a description above that is not explained, e.g. 25D+N

Figure S5 – This Figure is not referenced in the main text, either omit it or reference it. Text is indented unlike all other Figure description text.

---

## Author Response (AR2)

We once again thank the reviewers for their careful consideration of our paper. We have taken their recommendations into account and have altered the manuscript accordingly. The main concern cited by the reviewers was the use of surface area normalized $INP_{SSA}$ concentrations. We have thus changed the main text to show $INP_{SSA}$ normalized by SSA particle count, rather than surface area. We have added the following explanation to line 284:

*"Comparison of the total CPC-based SSA number concentration to the SSA number concentration derived from the DMPS revealed near unity, indicating nearly all of the particles number concentrations were captured by the DMPS. While studies typically present INP concentrations normalized by total SSA surface area, this was not possible in our experiment as the size distribution of supermicron particles was not monitored. However, in the supporting information, we do present a theoretical surface area normalized $INP_{SSA}$ calculation for comparison with other studies. The theoretical distribution was based on in-situ particle number concentration measurements at Mace Head and open-ocean eddy correlation flux measurements from the Eastern Atlantic (Table S1) (Ovadnevaite et al., 2014), with the resulting surface area distribution shown in Figure S1."*

We have thus altered Figure 1, the correlations in Tables 2 and 3 and their associated scatter plots, to show $INP_{SSA}$ normalized by SSA particles number concentrations. Furthermore, we altered the single component model in section 4 to account for the $INP_{SSA}$ normalized by particles number concentrations. Conclusions remain unchanged.

Since we still believe the theoretical calculation of surface area normalized INPSSA concentrations is useful, we have moved the comparison of concentrations with literature values in Figure 3 as well as comparison with MC18 model in Section 4 to the Supplementary Information.

The other concern by reviewer 2 related to DFPC processing volume:

*'With regard to the DFPC, I didn't notice any mention of volumes processed, which given the need to limit particle loading with the DFPC, and the low concentrations of IN in seawater, is needed.'*

We have thus added the following text to the manuscript on line 171 of the main text:

*'The volume sampled on each filter averaged $8.95x10^3 \pm 2.26$ $m^3$ of air.'*

[revised manuscript text omitted]